# Towards Deepening Graph Neural Networks: A GNTK-based Optimization Perspective

**Wei Huang** [*†]
University of Technology Sydney
`weihuang.uts@gmail.com`

**Yayong Li** [*]
University of Technology Sydney
`yayong.li@student.uts.edu.au`

**Weitao Du**
Northeastern University
`weitao.du@northwestern.edu`

**Jie Yin**
The University of Sydney
`jie.yin@sydney.edu.au`

**Richard Yi Da Xu & Ling Chen**
University of Technology Sydney
`{YiDa.Xu,ling.chen}@uts.edu.au`

**Miao Zhang**
Aalborg University
`miaoz@cs.aau.dk`

## Abstract

Graph convolutional networks (GCNs) and their variants have achieved great success in dealing with graph-structured data. Nevertheless, it is well known that deep GCNs suffer from the over-smoothing problem, where node representations tend to be indistinguishable as more layers are stacked up. The theoretical research to date on deep GCNs has focused primarily on expressive power rather than trainability, an optimization perspective. Compared to expressivity, trainability attempts to address a more fundamental question: Given a sufficiently expressive space of models, can we successfully find a good solution via gradient descent-based optimizers? This work fills this gap by exploiting the Graph Neural Tangent Kernel (GNTK), which governs the optimization trajectory under gradient descent for wide GCNs. We formulate the asymptotic behaviors of GNTK in the large depth, which enables us to reveal the dropping trainability of wide and deep GCNs at an exponential rate in the optimization process. Additionally, we extend our theoretical framework to analyze residual connection-based techniques, which are found to be merely able to mitigate the exponential decay of trainability mildly. Inspired by our theoretical insights on trainability, we propose Critical DropEdge, a connectivity-aware and graph-adaptive sampling method, to alleviate the exponential decay problem more fundamentally. Experimental evaluation consistently confirms using our proposed method can achieve better results compared to relevant counterparts with both infinite-width and finite-width.

## 1 Introduction

Recently, Graph Neural Networks (GNNs) have shown incredible abilities to learn node or graph representations and achieved superior performance on various downstream tasks, such as node classification (Kipf & Welling, 2017; Veličković et al., 2018; Hamilton et al., 2017), graph classification (Xu et al., 2019; Lee et al., 2019b; Yuan & Ji, 2020), and link prediction (Kipf & Welling, 2016), etc. However, most GNNs (e.g., GCNs) achieve their best only with a shallow depth, e.g., 2 or 3 layers, and their performance on those tasks would promptly degrade as the number of layers grows. Towards this phenomenon, research attempts have been made to deepen understanding of current GNN architectures and their expressive power. Li et al. (2018) showed that GCN is a special form of Laplacian smoothing, which mixes node representations with nearby neighbors. This mechanism potentially poses the risk of *over-smoothing* as more layers are stacked together, where node representations tend to be indistinguishable from each other. Oono & Suzuki (2020) investigated

---

[*]Equal Contribution.
[†]Work partially performed while at The University of Sydney

the expressive power of GNNs using the asymptotic behaviors as the layer goes to infinity. They proved that under certain conditions, the expressive power of GCN is determined by the topological information of the underlying graph inherent in the graph spectra.

Nevertheless, it remains elusive how to theoretically understand why deep GCNs fail to optimize. Existing theoretical investigation (Oono & Suzuki, 2020; Xu et al., 2019) on GNNs focus mainly on expressivity, which measures the complexity of functions that can be represented by a neural network. Exploring expressivity is theoretically convenient, but the corresponding conditions may be violated during the gradient training process, thereby leading to inconsistencies between theoretical conclusions and empirical results of trained networks (Gühring et al., 2020). Compared to expressivity, trainability addresses a more difficult but fundamental perspective of neural networks: How effectively a neural network can be optimized via gradient descent. The advantage of investigating trainability is that we can directly determine whether GNNs can be successfully trained under certain conditions, and to what extent. We are therefore inspired to raise two important questions:

- *Can we theoretically characterize the trainability of graph neural networks with respect to depth, thus understanding why deep GCNs fail to generalize?*

- *Can we further design an algorithm to facilitate deeper GCNs, benefiting from our theoretical investigation?*

Our answers are yes to both questions. We resort to the infinitely-wide multi-layer GCN to derive our solution. The research on infinitely-wide networks can be traced back to the seminal work of Neal (1996), which showed that single hidden layer networks with random weights at initialization (without training) are Gaussian Processes (GPs) in the infinite width limit. Later, the connection between GPs and multi-layer infinitely-wide networks with Gaussian initialization (Lee et al., 2018; de G. Matthews et al., 2018) and orthogonal weights (Huang et al., 2021) was reported. Recent trends in Neural Tangent Kernel (NTK) have led to a proliferation of studies on the optimization and generalization of infinitely (ultra)-wide networks. In particular, Jacot et al. (2018) made a groundbreaking discovery that gradient descent training in the infinite width limit can be captured by an NTK. Du et al. (2019b) formulated Graph Neural Tangent Kernel (GNTK) for infinitely-wide GNNs and shed light on theoretical guarantees for GNNs. Prior to the discovery of GNTK, there was little understanding of the non-convexity of GNNs, which is analytically intractable. In the learning regime of GCN governed by GNTK, the optimization becomes an almost convex problem, making GNTK a promising perspective to study the trainability of deep GCNs.

In this work, we leverage the GNTK techniques of infinitely-wide networks to investigate whether ultra-wide GCNs are trainable in the large depth. In particular, we formulate the large-depth asymptotic behavior of the GNTK, illuminated by innovative works on deep networks (Hayou et al., 2019b; Xiao et al., 2020), through which we can analyze the optimization properties of deep GCNs. Specifically, we make the following contributions:

- To our best knowledge, we are the first to investigate the trainability of deep GCNs through GNTK. We prove that all entries of a GNTK matrix regarding a pair of graphs converge exponentially to the same value, making the GNTK matrix singular in the large depth. We thus establish a corollary that the trainability of ultra-wide GCNs exponentially collapses on node classification tasks.

- We apply our theoretical analysis to the residual connection-based techniques for GCNs. Our theory shows that residual connection can, to some extent, slow down the exponential decay rate of trainability, but lack the ability to fundamentally solve the problem. This result enables to better understand why and to what extent recent residual connection-based methods work.

- Our theoretical framework provides insights to guide the development of deep GCNs. We further propose an edge-based sampling method, named Critical DropEdge, to effectively mitigate the exponential decay of trainability. This graph-adaptive and connectivity-aware method is easy to implement in both finitely-wide and infinitely-wide GNNs. Our experiments show using the proposed method can outperform competitors in the large depth.

## 2 BACKGROUND AND PRELIMINARIES

We first review the results of infinitely-wide neural networks at initialization. We then review NTK, making a connection to trainability. Finally, we introduce GCNs along with our setup and notation.

## 2.1 INFINITELY-WIDE NETWORKS AT INITIALIZATION

We begin by considering a fully-connected network of depth $L$ with width $m_l$ in each layer. The weight and bias in the $l$-th layer are denoted by $W^{(l)} \in \mathbb{R}^{m_l \times m_{l-1}}$ and $b^{(l)} \in \mathbb{R}^{m_l}$. Letting the pre-activations be given by $h_i^{(l)}$, information propagation in this network is governed by,

$$h_i^{(l)} = \frac{\sigma_w}{\sqrt{m_l}} \sum_{j=1}^{m_l} \phi(W_{ij}^{(l)} h_j^{(l-1)}) + \sigma_b b_i^{(l)} \tag{1}$$

where $\phi : \mathbb{R} \to \mathbb{R}$ is the activation function, $\sigma_w$ and $\sigma_b$ define the variance scale of the weights and biases, respectively. Given the parameterization that weights and biases are randomly generated by i.i.d. normal distribution, i.e., $W_{ij}^{(l)}, b_i^{(l)} \sim \mathcal{N}(0,1)$, the pre-activations are Gaussian distributed in the infinite width limit as $m_1, m_2, \ldots, m_{l-1} \to \infty$. This results from the central limit theorem (CLT). Consider a dataset $X \in \mathbb{R}^{n \times d}$ of size $n = |X|$, the covariance matrix of Gaussian process kernel (GPK) regarding infinitely-wide network is defined by $\Sigma^{(l)}(x, x') = \mathbb{E}[h_i^{(l)}(x) h_i^{(l)}(x')]$. According to the signal propagation (1), the covariance matrix or GPK with respect to layer can be described by a recursion relation, $\Sigma^{(l)}(x, x') = \sigma_w^2 \mathbb{E}_{h \sim \mathcal{N}(0, \Sigma^{(l-1)})}[\phi(h(x)) \phi(h(x'))] + \sigma_b^2$.

The mean-field theory is a paradigm that studies the limiting behavior of GPK, which is a measure of expressivity for networks (Poole et al., 2016; Schoenholz et al., 2017). In particular, expressivity describes to what extent two different inputs can be distinguished. The property of evolution for expressivity $\Sigma^{(l)}(x, x')$ is determined by how fast it converges to its fixed point $\Sigma^*(x, x') \equiv \lim_{l \to \infty} \Sigma^{(l)}(x, x')$. It is shown that in almost the entire parameter space spanned by hyper-parameters $\sigma_w$ and $\sigma_b$, the evolution exhibits a dramatic convergence rate formulated by an exponential function except for a critical line known as the *edge of chaos* (Poole et al., 2016; Schoenholz et al., 2017). Consequently, an infinitely-wide network loses its expressivity exponentially in most cases while retaining the expressivity at the edge of chaos. Given this reason, we focus on the edge of chaos in this work. In particular, we set the value of hyper-parameters to satisfy, $\sigma_w^2 \int \mathcal{D}_z [\phi'(\sqrt{q^*} z)]^2 = 1$, where $q^*$ is the fixed point of diagonal entries in the covariance matrix, and $\int \mathcal{D}z = \frac{1}{\sqrt{2\pi}} \int dz e^{-\frac{1}{2}z^2}$ is the measure for a normal distribution. For the ReLU activation, edge of chaos requires $\sigma_w^2 = 2$ and $\sigma_b^2 = 0$.

## 2.2 NEURAL TANGENT KERNEL AND TRAINABILITY

Most studies on infinitely-wide networks through mean-field theory (Poole et al., 2016; Schoenholz et al., 2017) have focused solely on initialization without training. Jacot et al. (2018) took a step further by considering infinitely-wide networks trained with gradient descent. Let $\eta$ be the learning rate, and $\mathcal{L}$ be the loss function. The dynamics of gradient flow for parameters $\theta = \text{vec}(\{W_{ij}^{(l)}, b_i^{(l)}\}) \in \mathbb{R}^{(\sum_l m_l(m_{l-1}+1)) \times 1}$, the vector of all parameters, is given by,

$$\frac{\partial \theta}{\partial t} = -\eta \nabla_\theta \mathcal{L} = -\eta \nabla_\theta f_t(X)^T \nabla_{f_t(X)} \mathcal{L} \tag{2}$$

Then, the dynamics of output functions $f(X) = \text{vec}(f(x)_{x \in X}) \in \mathbb{R}^{nm_L \times 1}$ follow,

$$\frac{\partial f_t(X)}{\partial t} = \nabla_\theta f_t(X) \frac{\partial \theta}{\partial t} = -\eta \Theta_t(X, X) \nabla_{f_t(X)} \mathcal{L} \tag{3}$$

where the NTK at time $t$ is defined as,

$$\Theta_t(X, X) \equiv \nabla_\theta f_t(X) \nabla_\theta f_t(X)^T \in \mathbb{R}^{nm_L \times nm_L} \tag{4}$$

In a general case, the NTK varies with the training time, thus providing no substantial insights into the convergence property of neural networks. Interestingly, as shown by Jacot et al. (2018), the NTK converges to an explicit limiting kernel and does not change during training in the infinite-width limit. This leads to a simple but profound result in the case of mean squared error (MSE) loss, $\mathcal{L} = \frac{1}{2} \|f_t(X) - Y\|_2^2$, where $Y$ is the label associated with the input $X$,

$$f_t(X) = (I - e^{-\eta \Theta_\infty(X,X)t}) Y + e^{-\eta \Theta_\infty(X,X)t} f_0(X) \tag{5}$$

where $\Theta_\infty$ is the limiting kernel. This is the solution to an ordinary differential equation. As the training time $t$ tends to infinity, the output function fits the label very well, i.e., $f_\infty(X) = Y$. As

proved by Lemma 1 in Hayou et al. (2019b), the network is trainable only if $\Theta_\infty(X, X)$ is non-singular. Quantitatively, the condition number $\kappa \equiv \lambda_{\max}/\lambda_{\min}$ can be a measure of trainability as confirmed by Xiao et al. (2020).

## 2.3 GRAPH CONVOLUTIONAL NETWORKS

We define an undirected graph as $G = (\mathcal{V}, \mathcal{E})$, where $\mathcal{V}$ is a set of nodes and $\mathcal{E}$ is a set of edges. We denote the number of nodes in graph $G$ by $n = |\mathcal{V}|$. The nodes are associated with a node feature matrix $X \in \mathbb{R}^{n \times d}$, and the corresponding labels are $Y \in \mathbb{R}^{n \times k}$, with $d$ and $k$ being the dimension of node features and number of classes, respectively. In this work, we develop our theory towards understanding the trainability of GCNs on node classification tasks.

GCNs iteratively update node features through aggregating and transforming the representations of their neighbors. Figure 3 in Appendix A illustrates an overview of the information propagation in a general GCN. We define a propagation unit to be the combination of a $R$-layer multi-layer perceptron (MLP) and one aggregation operation. We use subscript $(r)$ to denote the layer index of MLP in each propagation unit and superscript $(l)$ to indicate the index of aggregation operation, which is also the index of the propagation unit. $L$ is the total number of propagation units. To be specific, the node representation propagation in GCNs through an MLP follows the expression,

$$h_{(0)}^{(l)}(u) = \frac{1}{|\mathcal{N}(u)| + 1} \sum_{v \in \mathcal{N}(u) \cup u} h_{(R)}^{(l-1)}(v) \tag{6}$$

$$h_{(r)}^{(l)}(u) = \frac{\sigma_w}{\sqrt{m}} \phi(W_{(r)}^{(l)} h_{(r-1)}^{(l)}(u)) + \sigma_b b_{(r)}^{(l)} \tag{7}$$

where $h_{(0)}^{(0)} = X$, $W_{(r)}^{(l)} \in \mathbb{R}^{m_l \times m_{l-1}}$, and $b_{(r)}^{(l)} \in \mathbb{R}^{m_l}$ are the learnable weights and biases, respectively, $\phi$ is the activation function, $\mathcal{N}(u)$ is the neighborhood of node $u$, and $\mathcal{N}(u) \cup u$ is the union of node $u$ and its neighbors. Equation (6) reveals the node feature aggregation operation among its neighborhood according to a GCN variant (Hamilton et al., 2017). Equation (7) is a standard non-linear transformation with NTK-parameterization (Jacot et al., 2018), where $m$ is the width, i.e., number of neurons in each layer, $\sigma_w$ and $\sigma_b$ define the variance scale of the weights and biases. For the activation function, we focus on both ReLU and Tanh, which are denoted as $\phi(x) = \max\{0, x\}$ and $\phi(x) = \tanh(x)$, respectively. Without loss of generality, our theoretical framework can handle other common activation functions, whereas the GNTK work (Du et al., 2019b) only adopted ReLU.

# 3 AGGREGATION PROVABLY LEADS TO EXPONENTIAL TRAINABILITY LOSS

## 3.1 GNTK FORMULATION

Based on the definition of NTK (4), we recursively formulate the propagation of GNTK in the infinite-width limit. As information propagation in a GCN is built on two operations: aggregation (6) and non-linear transformation (7), the corresponding formulas of GNTK are expressed as follows,

$$\Theta_{(0)}^{(l)}(u, u') = \frac{1}{|\mathcal{N}(u)| + 1} \frac{1}{|\mathcal{N}(u')| + 1} \sum_{v \in \mathcal{N}(u) \cup u} \sum_{v' \in \mathcal{N}(u') \cup u'} \Theta_{(R)}^{(l-1)}(v, v') \tag{8}$$

$$\Theta_{(r)}^{(l)}(u, u') = \Theta_{(r-1)}^{(l)}(u, u') \dot{\Sigma}_{(r)}^{(l)}(u, u') + \Sigma_{(r)}^{(l)}(u, u') \tag{9}$$

The two equations above correspond to the aggregation operation and MLP transformation, respectively. To compute the GNTK with respect to the depth, the key step is to obtain the covariance matrix $\Sigma_{(r)}^{(l)}(u, u') \equiv \mathbb{E}[h_{(r)}^{(l)}(u) h_{(r)}^{(l)}(u')]$. According to the CLT, node representation $h_{(r)}^{(l)}(u)$ is a Gaussian distribution in the infinite-width limit. Applying this result to equations (6) and (7), the resultant covariance matrix is composed of two parts,

$$\Sigma_{(0)}^{(l)}(u, u') = \frac{1}{|\mathcal{N}(u)| + 1} \frac{1}{|\mathcal{N}(u')| + 1} \sum_{v \in \mathcal{N}(u) \cup u} \sum_{v' \in \mathcal{N}(u') \cup u'} \Sigma_{(R)}^{(l-1)}(v, v') \tag{10}$$

$$\Sigma_{(r)}^{(l)}(u, u') = \sigma_w^2 \mathbb{E}_{z_1, z_2 \sim \mathcal{N}\left(0, \bar{\Sigma}_{(r-1)}^{(l)}\right)} \left[\phi(z_1)\phi(z_2)\right] + \sigma_b^2$$

$$\dot{\Sigma}_{(r)}^{(l)}(u, u') = \sigma_w^2 \mathbb{E}_{z_1, z_2 \sim \mathcal{N}\left(0, \bar{\Sigma}_{(r-1)}^{(l)}\right)} \left[\dot{\phi}(z_1)\dot{\phi}(z_2)\right] + \sigma_b^2 \tag{11}$$

The first equation (10) results from the aggregation operation. Meanwhile, the second equation (11) corresponds to the $R$-times non-linear transformations, where $\mathbb{E}_{z_1,z_2}$ takes the expectation with respect to a centered Gaussian process of covariance $\tilde{\Sigma}^{(l)}_{(r-1)} \in \mathbb{R}^{2\times2}$ for previous MLP layer across $u, u'$, and $\dot{\phi}$ denotes the derivative of $\phi$.

## 3.2 Trainability in the Large Depth

We aim to characterize the behavior of GNTK matrix $\Theta^{(l)}_{(r)}(G) \in \mathbb{R}^{n\times n}$, as the depth tends to infinity. From the GNTK formulation, both aggregation (8) and transformation (9) contribute simultaneously to the final limiting result. We derive our theorem on the asymptotic behavior of infinitely-wide GCN in the large depth limit, which is given as follows:

**Theorem 1** (Convergence rate of GNTK). *If transition matrix $\mathcal{A}(G) \in \mathbb{R}^{n^2\times n^2}$ is irreducible and aperiodic, with a stationary distribution vector $\vec{\pi}(G) \in \mathbb{R}^{n^2\times 1}$, where $\vec{\Theta}^{(l)}_{(0)}(G) = \mathcal{A}(G)\vec{\Theta}^{(l-1)}_{(R)}(G)$ and $\vec{\Theta}^{(l)}_{(r)}(G) \in \mathbb{R}^{n^2\times 1}$ is the result of being vectorized. Then, there exist constants $0 < \alpha < 1$ and $C > 0$, and constant vectors $\vec{v}, \vec{v}' \in \mathbb{R}^{n^2\times 1}$ depending on the number of MLP iterations $R$, such that $\left| \Theta^{(l)}_{(r)}(u, u') - \vec{\pi}(G)^T\left(Rl\vec{v} + \vec{v}'\right) \right| \le C\alpha^l$.*

The proof sketch follows a divide-and-conquer manner. In particular, we first analyze the network with only aggregation and prove that $\mathcal{A}(G)$ is a Markov transition matrix. Then, we formulate the behavior of MLP in the large depth based on Hayou et al. (2019b). Finally, we derive the final result by considering the two operations. We leave the complete proof in Appendix B.

In Theorem 1, we rigorously characterize the convergence properties of GNTK in the large depth limit. As the depth goes to infinity, all entries in the GNTK converge to a unique quantity at an exponential rate. We thus have $\Theta^{(l)}_{(r)}(G) \approx \vec{\pi}(G)^T(Rl\vec{v})\mathbf{1}_{n\times n}$ as $l \to \infty$, where $\mathbf{1}_{n\times n}$ is an $(n \times n)$-dimensional matrix whose entries are one. The exponential convergence rate of $\Theta^{(l)}_{(r)}(G)$ implies that the trainability of infinitely-wide GCNs degenerates dramatically, as stated below.

**Corollary 1** (Trainability of ultra-wide GCNs). *Consider a GCN of the form (6) and (7), with depth $l$, number of non-linear transformations $r$, an MSE loss, and a Lipchitz activation, trained with gradient descent on a node classification task. Then, the output function follows, $f_t(X) = e^{-\eta\Theta^{(l)}_{(r)}(G)t}f_0(X) + (I - e^{-\eta\Theta^{(l)}_{(r)}(G)t})Y$. Then, $\Theta^{(l)}_{(r)}(G)$ is singular when $l \to \infty$. Moreover, there exists a constant $C > 0$ such that for all $t > 0$, $\|f_t(X) - Y\| > C$.*

We leave proof in Appendix C. According to the above corollary, as $l \to \infty$, the GNTK matrix would become a singular matrix. This would lead to a discrepancy between output $f_t(X)$ and label $Y$, which means GNTK loses the ability to fit the label. Therefore, an ultra-wide GCN with a large depth cannot be trained successfully on node classification tasks.

## 4 Towards Deepening Graph Neural Networks

### 4.1 Theoretical Analysis on Residual Connection

We have so far characterized the trainability of vanilla GCNs through the GNTK and showed that the trainability of ultra-wide GCNs drops at an exponential rate. Recently, considerable efforts have been made to deepen GCNs, among which residual connection-based techniques are widely applied to resolve the over-smoothing problem (Li et al., 2019). We now apply our theoretical framework to analyze to what extent residual connection techniques could alleviate the trainability loss problem.

We first consider residual connection in aggregation, in which the propagation of the GNTK can be formulated as,

$$\vec{\Theta}^{(l)}(G) = (1 - \delta)\mathcal{A}(G)\vec{\Theta}^{(l-1)}(G) + \delta\vec{\Theta}^{(l-1)}(G), \tag{12}$$

where $0 < \delta < 1$. Taking equation (12) as a new aggregation process, then $\vec{\Theta}^{(l)}(G) = \tilde{\mathcal{A}}(G)\vec{\Theta}^{(l-1)}(G)$, where $\tilde{\mathcal{A}}(G) = (1 - \delta)\mathcal{A}(G) + \delta I$. We prove that $\tilde{\mathcal{A}}(G)$ is also a transition matrix with a greater second largest eigenvalue compared to the original matrix $\mathcal{A}(G)$.

**Theorem 2** (Convergence rate of residual connection in aggregation). *Consider a GNTK of non-linear transformation (9) and residual connection (12). Then with a stationary vector $\tilde{\vec{\pi}}(G)$ for $\tilde{\mathcal{A}}(G)$, there exist constants $0 < \tilde{\alpha} < 1$ and $C > 0$, and constant vectors $\vec{v}$ and $\vec{v}'$ depending on $R$, such that $\left| \Theta_{(r)}^{(l)}(u, u') - \tilde{\vec{\pi}}(G)^T \left( Rl\vec{v} + \vec{v}' \right) \right| \le C\tilde{\alpha}^l$. Furthermore, we denote the second largest eigenvalue of $\mathcal{A}(G)$ and $\tilde{\mathcal{A}}(G)$ as $\lambda_2$ and $\tilde{\lambda}_2$, respectively. Then, $\tilde{\lambda}_2 > \lambda_2$.*

The detailed proof of Theorem 2 and the relationship between convergence rate and the second largest eigvenvalue can be found in Appendix D.1. This theorem implies that a residual connection in aggregation can slow down the convergence rate, which is consistent with empirical observations that residual connection can help deepen GCNs.

Then, we consider residual connection only applied on non-linear transformation (MLP). In this case, the recursive equation for the corresponding GNTK can be expressed as,

$$\Theta_{(r)}^{(l)}(u, u') = \Theta_{(r-1)}^{(l)}(u, u')\left(\dot{\Sigma}_{(r)}^{(l)}(u, u') + 1\right) + \Sigma_{(r)}^{(l)}(u, u') \tag{13}$$

This formula is similar to the vanilla GNTK in the infinite-width limit. Only an additional residual term appears according to residual connection. It turns out that this term may not help slow down the convergence rate for non-linear transformation.

**Theorem 3** (Convergence rate of GNTK with residual connection between transformations). *Consider a GNTK of the form (8) and (13). If $\mathcal{A}(G)$ is irreducible and aperiodic, with a stationary distribution $\vec{\pi}(G)$, then there exist constants $0 < \alpha < 1$ and $C > 0$, and constant vectors $\vec{v}$ and $\vec{v}'$ depending on $R$, such that, $\left| \Theta_{(r)}^{(l)}(u, u') - \vec{\pi}(G)^T \left( Rl(1 + \frac{\sigma_w^2}{2})^{Rl}\vec{v} + \vec{v}' \right) \right| \le C\alpha^l$.*

Note that $\alpha$ in Theorem 3 is the same as in Theorem 1. The proof of Theorem 3 can be found in Appendix D.2. Theorem 3 demonstrates that adding residual connection in MLP can not even reduce the convergence rate of trainability. Finally, we consider residual connection applied to both aggregation and non-linear transformation simultaneously:

**Corollary 2** (Convergence rate of GNTK with residual connection in aggregation and transformation). *Consider a GNTK of the form (12) and (13). If $\tilde{\mathcal{A}}(G)$ is irreducible and aperiodic, with a stationary distribution $\tilde{\vec{\pi}}(G)$, there exist constants $0 < \tilde{\alpha} < 1$ and $C > 0$, and constant vectors $\vec{v}, \vec{v}' \in \mathbb{R}^{n^2 \times 1}$ depending on $R$, such that $\left| \Theta_{(r)}^{(l)}(u, u') - \tilde{\vec{\pi}}(G)^T \left( Rl(1 + \frac{\sigma_w^2}{2})^{Rl}\vec{v} + \vec{v}' \right) \right| \le C\tilde{\alpha}^l$.*

## 4.2 A NEW SAMPLING METHOD: CRITICAL DROPEDGE

Residual connection is designed from a layer-wise perspective, but it has limited abilities to mitigate the exponential decay of trainability. To better resolve this problem, we need to look deeper into the root cause of the problem – the transition matrix corresponding to the aggregation operation. A necessary condition for matrix $\mathcal{A}(G)$ to be a probability transition matrix is that graph $G$ is connected. Thus, breaking the connectivity condition is a promising way of better solving the exponential decay problem. One such method is to perform edge sampling guided by the critical percolation theory (Huang et al., 2018; Erdős & Rényi, 1961) in random graphs.

On a finite complete graph of $n$ nodes, there exist $E_t = n(n - 1)/2$ edges between all pairs of nodes. A random graph $\hat{G}$ is achieved by randomly and uniformly preserving some edges from the complete graph with an edge probability as $p = \frac{|E|}{E_t}$, where $|E|$ is the number of edges preserved in the random graph. In this way, the critical percolation can be realized in the random graph with a critical edge probability $p_c = 1/(n - 1)$ (Erdős & Rényi, 1961). In the thermodynamic limit of $n \to \infty$, the critical random graph exhibits critical connectivity: the probability that there exists a path from a fixed point to another point within a certain distance decreases polynomially.

**Proposition 1** (Critical connectivity in random graph (Erdős & Rényi, 1961)). *Suppose a random graph $\hat{G}$ has $n$ nodes with a constant edge probability $p$. (1) If $p < p_c$, then almost every random graph is such that its largest component[1] is of size $O(\log n)$; (2) If $p > p_c$, the random graph has a giant component of size $(1 - \alpha_p + o(1))n$, where $\alpha_p < 1$; (3) If $p = p_c$, then the maximal size of a component of almost every graph has order $n^{2/3}$.*

---

[1] In graph theory, a component of an undirected graph is an induced subgraph in which any two nodes are connected to each other by paths.

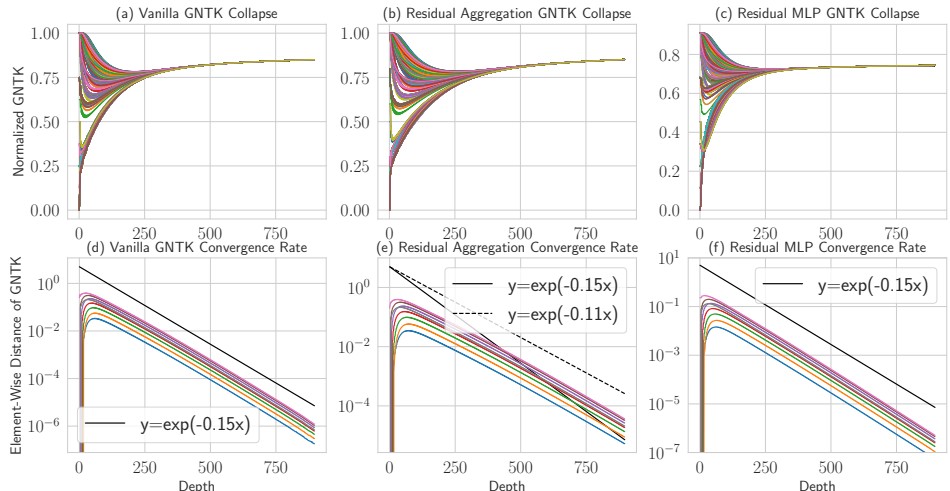

Figure 1: Convergence rate of GNKT. (a)-(c) Entries of the normalized (residual connection) GNTK as a function of the depth, defined as $Rl + r$. All entries tend to have the same value as the depth grows. (d)-(f) The element (entry)-wise distance of the normalized (residual connection) GNTK as a function of the depth. The convergence rate can be bounded by an exponential function $y = \exp(-0.15x)$ for vanilla and residual MLP GNTK, whereas the convergence rate of residual aggregation is bounded by $y = \exp(-0.11x)$.

The proposition above implies that the information transforms in the critical random graph at a polynomial rate rather than an exponential rate. This inspires us to solve the problem of exponentially dropping trainability through designing a graph-dependent and connectivity-aware sampling algorithm called Critical DropEdge. In particular, given a graph $G$, we randomly drop some edges and preserve the number of edges as $E_r = E_t \cdot p_c = n/2$, to approximate the critical graph. Unlike DropEdge (Rong et al., 2019) that randomly removes a certain number of edges from the input graph, our method fixes the edge preserving percentage as $\rho = E_r/|E| = \frac{|V|}{2|E|}$. It is worth noting that DropEdge may choose the edge probability as $p < p_c$ where information can only be passed to a distance of $O(\log n)$ in the graph, or $p > p_c$ where the exponential decay of trainability may occur. A further discussion of the relationship between Proposition 1 and the trainability of the corresponding GNTK can be found in Appendix E.

## 5 EXPERIMENTS

In this section, we empirically verify our theoretical results and validate the proposed Critical DropEdge method on node classification tasks. Details of four real-world graph datasets used for node classification are summarized in Table 3 in Appendix F.1.

### 5.1 CONVERGENCE RESULTS OF GNTKS

Theorems 1-3 provide theoretical convergence rates for (residual) GNTK. We show the corresponding numerical verification in Figure 1. We select a graph randomly from a bioinformatics dataset (i.e., MUTAG), which consists of 18 nodes and 21 edges. We generate a GNTK of the graph using the implementation of Du et al. (2019b), with ReLU activation, $R = 3$, and $L = 300$. Figure 1(a)-(c) show that all entries of normalized GNTKs converge to an identical value as the depth goes larger. Figure 1(d)-(f) further indicates that the convergence rate of GNTK is exponential, as reflected by our theorems. By comparing convergence rates, we conclude that the residual connection in aggregation can slow down the convergence rate, which is consistent with Theorem 2.

### 5.2 TRAINABILITY OF WIDE GCNS

We further examine whether ultra-wide GCNs can be trained successfully for node classification. We conduct experiments on a GCN (Kipf & Welling, 2017), where we apply a width of $1,000$ at

each hidden layer and the depth ranging from 2 to 29. Figure 2 shows the training and test accuracy on Cora, Citesser and Pubmed after 300 training epochs. These results show a dramatic drop in both training and test accuracy as the depth grows, confirming that wide GCNs lose trainability significantly in the large depth on node classification, as revealed by Corollary 1.

### 5.3 Performance of Critical DropEdge

We apply Critical DropEdge (referred to as C-DropEdge) to finitely-wide and infinitely-wide GNNs on semi-supervised node classification. The implementation details are given in Appendix F.2.

For finitely-wide GNNs, we consider three back-bones: GCN, JKNet, and IncepGCN (Rong et al., 2019). For DropEdge, we use the hyper-parameters reported in Rong et al. (2019) to obtain the results. For C-DropEdge, we perform a random hyper-parameter search and fix the edge preserving rate as $\rho(G) = \frac{|V|}{2|E|}$. In addition, we compare with DGN (Zhou et al., 2020), a normalization-based baseline, which uses GCN (Kipf & Welling, 2017) and GAT (Veličković et al., 2018) as backbones.

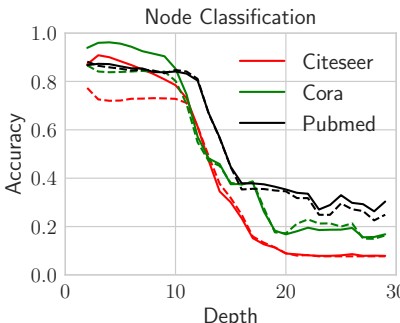

Figure 2: Training and test accuracy w.r.t. model depth. Solid and dashed lines are train and test accuracy respectively.

Table 1 summarizes node classification performance of finitely-wide GNNs with 4/8/16/32 layers on three citation networks (Cora, Citesser, Pubmed) and one co-author network (Physics). In particular, we report the best performance across different backbones for DropEdge, C-DropEdge and DGN, and leave separate results with different backbones in Appendix G.2. The reported results are the mean and standard deviation over 10 times. As can be seen, C-DropEdge consistently outperforms GCN, DGN, and DropEdge, especially when the model is deep. Besides, C-DropEdge can achieve smaller error variances, demonstrating stronger robustness than DropEdge.

For infinitely-wide GCNs, we consider two backbones: GCN (Kipf & Welling, 2017) and JKNet (Xu et al., 2018). The corresponding results can be found in Appendix G.1.

**Comparison to DropEdge.** DropEdge and C-DropEdge differ largely in their hyper-parameter search space. To ensure good performance, DropEdge needs to exhaustively search for the most appropriate edge preserving percentage – one of the most significant hyper-parameters – that has a crucial influence on the final performance. In contrast, C-DropEdge has a fixed and graph-dependent edge preserving rate, which implies its hyper-parameter space is much smaller than that of DropEdge. To verify if C-DropEdge can achieve the results close to optimal, we conduct experiments with various edge preserving rates, and present the results in Table 2. We conclude that, from both theoretical and empirical perspectives, the edge preserving percentage set by C-DropEdge is reasonable and effective, achieving the results close to optimal.

## 6 Related Work

**Neural Tangent Kernel.** NTKs are used to describe the dynamics of infinitely-wide networks during gradient descent training. In the infinite-width limit, NTK converges to an explicit limiting kernel; besides, it stays constant during training, providing a convergence guarantee for over-parameterized networks (Jacot et al., 2018; Lee et al., 2019a; Allen-Zhu et al., 2019; Du et al., 2019a; Zou et al., 2018). Besides, NTK has been applied to various architectures and brought a wealth of results, such as orthogonal initialization (Huang et al., 2021), convolutions (Arora et al., 2019), SVM (Chen et al., 2021), attention (Hron et al., 2020). As for graph networks, GNTK helps us understand how GNNs learn a class of smooth functions on graphs (Du et al., 2019b) and how they extrapolate differently from multi-layer perceptron (Xu et al., 2020).

**Deep Graph Neural Networks.** Since deep GNNs suffer from the over-smoothing problem, a large and growing body of literature has made efforts in deepening GNNs. There is a line of methods that resort to *residual connection* to retain their feature expressivity in deep layers. Xu et al. (2018) use skip connection along with node representations from different neighborhood ranges to preserve the locality of node representations. Klicpera et al. (2019) derive a personalized propagation of

Table 1: Comparison results of test accuracy (%) between C-DropEdge, GCN, DropEdge, and DGN.

| Datasets | Methods | 4-layer | 8-layer | 16-layer | 32-layer |
|---|---|---|---|---|---|
| Cora | GCN | $79.8 \pm 1.1$ | $73.2 \pm 2.7$ | $36.3 \pm 13.8$ | $20.1 \pm 2.4$ |
| | DropEdge | $82.2 \pm 0.7$ | $82.0 \pm 0.9$ | $82.2 \pm 0.7$ | $82.1 \pm 0.5$ |
| | DGN | $82.0 \pm 0.9$ | $80.2 \pm 0.8$ | $77.7 \pm 1.0$ | $73.0 \pm 0.8$ |
| | C-DropEdge | $\mathbf{82.5 \pm 0.7}$ | $\mathbf{82.3 \pm 0.6}$ | $\mathbf{82.4 \pm 0.8}$ | $\mathbf{82.6 \pm 0.9}$ |
| Citeseer | GCN | $61.2 \pm 3.0$ | $50.2 \pm 5.7$ | $30.8 \pm 2.2$ | $21.7 \pm 3.0$ |
| | DropEdge | $70.2 \pm 1.0$ | $70.8 \pm 1.1$ | $70.7 \pm 1.0$ | $70.2 \pm 0.8$ |
| | DGN | $69.0 \pm 0.9$ | $66.5 \pm 1.1$ | $62.9 \pm 1.2$ | $63.2 \pm 0.9$ |
| | C-DropEdge | $\mathbf{70.8 \pm 0.6}$ | $\mathbf{70.9 \pm 0.9}$ | $\mathbf{71.0 \pm 1.0}$ | $\mathbf{70.7 \pm 0.9}$ |
| Pubmed | GCN | $77.4 \pm 0.7$ | $57.2 \pm 8.4$ | $39.5 \pm 10.3$ | $36.3 \pm 8.4$ |
| | Dropedge | $77.6 \pm 1.4$ | $77.3 \pm 1.3$ | $76.7 \pm 1.3$ | $77.2 \pm 1.3$ |
| | DGN | $\mathbf{78.2 \pm 1.0}$ | $77.8 \pm 1.2$ | $77.2 \pm 1.3$ | $77.0 \pm 1.1$ |
| | C-DropEdge | $78.0 \pm 0.4$ | $\mathbf{77.9 \pm 1.0}$ | $\mathbf{77.2 \pm 1.0}$ | $\mathbf{77.8 \pm 1.0}$ |
| Physics | GCN | $90.2 \pm 0.9$ | $83.5 \pm 2.2$ | $41.6 \pm 6.2$ | $28.8 \pm 9.4$ |
| | Dropedge | $91.6 \pm 0.8$ | $91.5 \pm 0.7$ | $91.2 \pm 0.5$ | $91.3 \pm 0.8$ |
| | DGN | $\mathbf{92.2 \pm 1.0}$ | $86.4 \pm 0.7$ | $83.4 \pm 0.6$ | $83.2 \pm 0.8$ |
| | C-DropEdge | $91.9 \pm 0.7$ | $\mathbf{91.7 \pm 0.6}$ | $\mathbf{92.0 \pm 0.4}$ | $\mathbf{91.6 \pm 0.6}$ |

Table 2: Comparison results of test accuracy at various edge preserving rates. Critical preserving percentage for each dataset is marked in bold. The three best results per model are shaded in blue .

| Cora | | | Citeseer | | | Pubmed | | |
|---|---|---|---|---|---|---|---|---|
| Percentage | GCN-8 | JKNet-4 | Percentage | GCN-4 | IncepGCN-4 | Percentage | JKNet-16 | IncepGCN-32 |
| 0.05 | $58.2 \pm 19.6$ | $82.0 \pm 0.6$ | 0.15 | $68.8 \pm 1.2$ | $70.1 \pm 0.7$ | 0.01 | $76.1 \pm 1.8$ | $75.5 \pm 1.9$ |
| 0.10 | $69.6 \pm 14.4$ | $82.1 \pm 0.6$ | 0.20 | $68.8 \pm 1.1$ | $70.6 \pm 0.9$ | 0.05 | $76.2 \pm 1.5$ | $76.1 \pm 1.3$ |
| 0.15 | $69.7 \pm 12.5$ | $82.2 \pm 0.6$ | 0.25 | $68.7 \pm 0.8$ | $70.5 \pm 0.9$ | 0.10 | $76.0 \pm 1.4$ | $76.8 \pm 1.3$ |
| 0.20 | $75.4 \pm 4.0$ | $82.5 \pm 0.7$ | 0.30 | $68.9 \pm 0.8$ | $70.5 \pm 0.9$ | 0.15 | $76.9 \pm 0.9$ | $77.0 \pm 1.4$ |
| **0.25** | $77.3 \pm 2.5$ | $82.5 \pm 0.7$ | **0.35** | $69.0 \pm 0.8$ | $70.8 \pm 0.6$ | **0.22** | $76.9 \pm 0.9$ | $77.8 \pm 0.9$ |
| 0.30 | $77.2 \pm 2.7$ | $82.2 \pm 1.1$ | 0.40 | $68.9 \pm 0.9$ | $70.5 \pm 0.4$ | 0.25 | $76.9 \pm 1.1$ | $75.8 \pm 2.5$ |
| 0.35 | $77.2 \pm 2.8$ | $81.7 \pm 0.8$ | 0.45 | $68.4 \pm 1.7$ | $70.1 \pm 0.7$ | 0.30 | $76.8 \pm 1.1$ | $77.1 \pm 1.2$ |
| 0.40 | $74.9 \pm 5.7$ | $81.4 \pm 0.6$ | 0.50 | $68.1 \pm 0.9$ | $70.2 \pm 0.8$ | 0.35 | $76.4 \pm 1.2$ | $77.6 \pm 1.3$ |
| 0.45 | $75.5 \pm 6.4$ | $81.7 \pm 0.7$ | 0.55 | $68.0 \pm 1.0$ | $70.3 \pm 0.7$ | 0.40 | $76.3 \pm 1.3$ | $77.6 \pm 1.1$ |

neural predictions (PPNP) based on personalized Pagerank. Other similar works using residual connection can also be seen in (Li et al., 2019; Gong et al., 2020; Chen et al., 2020; Liu et al., 2020). Another line of approaches tackle the issue via regularization mechanisms, such as node/edge dropping (Hou et al., 2019; Rong et al., 2019), batch normalization (Dwivedi et al., 2020), pair normalization (Zhao & Akoglu, 2020), and group normalization (Zhou et al., 2020). Recently, a series of latest works (Loukas, 2020; Zeng et al., 2020; Li et al., 2020; Cong et al., 2021; Huang et al., 2020) explore the underlying reasons for performance degradation towards mitigation solutions.

## 7 CONCLUSION AND DISCUSSION

In this work, we have characterized the asymptotic behavior of GNTK to measure the trainability of wide GCNs in the large depth. We prove that the trainability drops at an exponential rate due to the aggregation operation. Furthermore, we apply our theoretical framework to investigate to what extent residual connection-based techniques help deepen GCNs. We demonstrate that these techniques can merely slow down the decay rate, but are unable to solve the exponential decay problem in essence. To overcome the trainability loss problem, we further propose Critical DropEdge illuminated by our theoretical framework. The experimental results confirm that our method can mitigate the trainability problem of deep GCNs. Future research directions include designing a critical node-centric method so as to make better use of node information.

## ACKNOWLEDGMENTS

This work was partially supported by a Collaborative Research Project grant between The University of Sydney and Data61, Australia. We thank the anonymous reviewers for useful suggestions to improve the paper. We also thank Ye Su for helpful discussions. This work was also in collaboration with Digital Research Centre Denmark – DIREC, supported by Innovation Fund Denmark.

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

## A APPENDIX: A GENERAL GCN ARCHITECTURE

GCNs iteratively update node features through aggregating and transforming the representation of their neighbors. Figure 3 illustrates an overview of information propagation in a general GCN considered in this work.

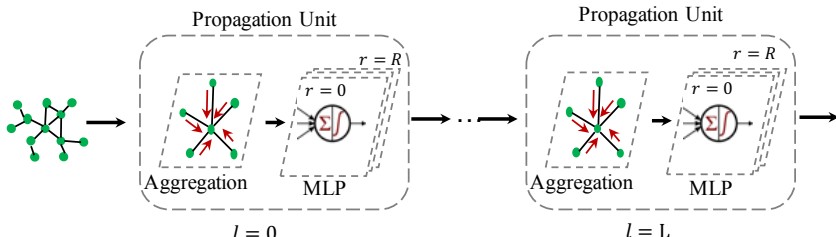

Figure 3: Overview of the information propagation in a general GCN.

## B APPENDIX: PROOFS FOR THEOREM 1

### B.1 CONVERGENCE OF AGGREGATION GNTK

In Theorem 1 we demonstrate that matrix $\mathcal{A}(G)$ is a transition matrix, we first prove this proposition. To this end, we block the non-linear transformation by setting $R = 0$. Then, the propagation of GNTK can be expressed as,

$$\Theta^{(l)}(u, u') = \frac{1}{|\mathcal{N}(u)| + 1} \frac{1}{|\mathcal{N}(u')| + 1} \sum_{v \in \mathcal{N}(u) \cup u} \sum_{v' \in \mathcal{N}(u') \cup u'} \Theta^{(l-1)}(v, v') \tag{14}$$

In order to facilitate the calculation, we rewrite Equation (14) as the following format,

$$\vec{\Theta}^{(l)}(G) = \mathcal{A}(G) \vec{\Theta}^{(l-1)}(G) \tag{15}$$

where $\vec{\Theta}^{(l)}(G) \in \mathbb{R}^{n^2 \times 1}$ is the result of being vectorized, and matrix $\mathcal{A}(G) \in \mathbb{R}^{n^2 \times n^2}$ is a square matrix. We show the key result that $\mathcal{A}(G)$ is a probability transition matrix, and the limiting behavior of $\Theta^{(l)}(G)$ in the following lemma,

**Lemma 1** (Convergence of aggregation). *Assume $R = 0$, then*

$$\lim_{l \to \infty} \Theta^{(l)}(u, u') = \vec{\pi}(G)^T \vec{\Theta}^{(0)}(G)$$

*where $\pi(G) \in \mathbb{R}^{n^2 \times 1}$, satisfying $\mathcal{A}(G)\vec{\pi}(G) = \vec{\pi}(G)$.*

*Proof.* When $R = 0$, Equation (14) reduces to,

$$\Theta^{(l)}(u, u') = \frac{1}{|\mathcal{N}(u)| + 1} \frac{1}{|\mathcal{N}(u')| + 1} \sum_{v \in \mathcal{N}(u) \cup u} \sum_{v' \in \mathcal{N}(u') \cup u'} \Theta^{(l-1)}(v, v')$$

In order to facilitate calculation, we rewrite the above equation in the format of matrix,

$$\vec{\Theta}^{(l)}(G) = \mathcal{A}^{(l)}(G) \vec{\Theta}^{(l-1)}(G)$$

where $\vec{\Theta}^{(l)}(G) \in \mathbb{R}^{n^2 \times 1}$, is the result of being vectorized. Thus, the matrix operation $\mathcal{A}^{(l)}(G) \in \mathbb{R}^{n^2 \times n^2}$. It is obvious that,

$$\mathcal{A}^{(L)}(G) = \mathcal{A}^{(L-1)}(G) = \cdots = \mathcal{A}^{(l)}(G) = \cdots = \mathcal{A}^{(1)}(G)$$

This implies that the aggregation operation is the same for each layer. The next step is to prove $\mathcal{A}(G)$ is a stochastic matrix (transition matrix):

$$\sum_j \mathcal{A}(G)_{ij} = 1.$$

According to Equation (14), $\mathcal{A}(G)$ can be expressed as a Kronecker product of two matrices,

$$\mathcal{A}(G) = \big[\mathcal{B}(G)\mathcal{C}(G)\big] \otimes \big[\mathcal{B}(G)\mathcal{C}(G)\big]$$

where $\mathcal{B}(G), \mathcal{C}(G) \in \mathbb{R}^{n \times n}$. We then analyse the two matrices separately:

(1) $\mathcal{B}(G)$ is a diagonal matrix, which corresponds to the factor $\frac{1}{\mathcal{N}(u)+1}$ .

$$\mathcal{B}(G) = \begin{pmatrix} \frac{1}{\mathcal{N}(u_1)+1} & & \\ & \ddots & \\ & & \frac{1}{\mathcal{N}(u_n)+1} \end{pmatrix}$$

(2) The element of matrix $\mathcal{C}(G)$ is determined by whether there exists an edge between two vertices,

$$\mathcal{C}(G)_{ij} = \tilde{\delta}_{ij}$$

where $\tilde{\delta}_{ij} = 1$ if $i == j$ or there is an edge between vertex $i$ and $j$, otherwise $\tilde{\delta}_{ij} = 0$.

We then use the property of matrix $\mathcal{B}$ and $\mathcal{C}$ before Kronecker product,

$$\sum_j \big[\mathcal{B}(G)\mathcal{C}(G)\big]_{ij} = \frac{1}{\mathcal{N}(u_i)+1} \sum_j \tilde{\delta}_{ij} = \frac{1}{\mathcal{N}(u_i)+1}(\mathcal{N}(u_i)+1) = 1$$

According to the definition of Kronecker product, we have,

$$\sum_j \mathcal{A}^{(l)}(G)_{ij} = \sum_b \sum_d [\mathcal{B}(G)\mathcal{C}(G)]_{ab}[\mathcal{B}(G)\mathcal{C}(G)]_{cd} = 1$$

where $i = a + (c-1)n$, and $j = b + (d-1)n$.

So far, we have proved that $\mathcal{A}(G)$ is a stochastic matrix. According to the Perron-Frobenius Theory, a stationary probability vector $\vec{\pi}(G) \in \mathbb{R}^{n^2 \times 1}$ is defined as a distribution, which does not change under application of the transition matrix; that is, it is defined as a probability distribution, which is also an eigenvector of the probability matrix, associated with eigenvalue 1:

$$\mathcal{A}(G)\vec{\pi}(G) = \vec{\pi}(G)$$

Note that the spectral radius of every stochastic matrix is at most 1 by Gershgorin circle theorem. Thus, the convergence rate is governed by the second largest eigenvalue.

Since $\lim_{l \to \infty} \mathcal{A}^{(l)}_{ij}(G) = \vec{\pi}_j(G)$, we have,

$$\lim_{l \to \infty} \vec{\Theta}^{(l)}(G) = \lim_{l \to \infty} \mathcal{A}^l(G)\vec{\Theta}^{(0)}(G) = \Pi(G)\vec{\Theta}^{(0)}(G)$$

where $\Pi(G) = \begin{pmatrix} \vec{\pi}(G)^T \\ \vec{\pi}(G)^T \\ \vdots \\ \vec{\pi}(G)^T \end{pmatrix}$ is the $n^2 \times n^2$ matrix all of whose rows are the stationary distribution. Then, we can see that every element in $\vec{\Theta}^{(l)}(G)$ converges exponentially to an identical value, depending on the stationary distribution and initial state,

$$\lim_{l \to \infty} \Theta^{(l)}(u, u') = \vec{\pi}(G)^T \vec{\Theta}^{(0)}(G)$$

$\square$

**Remark 1.** *This lemma can be extended to the multi-graph setting, where matrix $\mathcal{A}(G, G') \in \mathbb{R}^{nn' \times nn'}$ with respect to a pair of graphs $G, G'$ is a transition matrix as well, and $n'$ is the number of vertices in graph $G'$.*

### B.2 CONVERGENCE OF MLP GPK

Before we prove Theorem 1, we introduce the result for the Gaussian Process kernel $\Sigma_{(r)}^{(l)}(G)$ of a pure MLP. By doing so, we consider a network with only non-linear transformation, known as a pure MLP. This leads to $\Sigma_{(r)}^{(l)}(u, u') = \Sigma_{(r)}(u, u')$, where we use subscript $(r)$ to denote the layer index. And we rewrite the propagation function for GPK as follows,

$$\Sigma_{(r)}(u, u') = \sigma_w^2 \mathbb{E}_{z_1, z_2 \sim \mathcal{N}\left(0, \tilde{\Sigma}_{(r-1)}\right)} \left[\phi(z_1)\phi(z_2)\right] + \sigma_b^2 \tag{16}$$

and the variance $\tilde{\Sigma}_{(r)} \in \mathbb{R}^{2 \times 2}$ is,

$$\tilde{\Sigma}_{(r)} = \begin{pmatrix} \Sigma_{(r)}(u, u) & \Sigma_{(r)}(u, u') \\ \Sigma_{(r)}(u', u) & \Sigma_{(r)}(u', u') \end{pmatrix} \tag{17}$$

The large depth behavior has been well studied in Hayou et al. (2019a), and we introduce the result when the *edge of chaos* is realized. In particular, we set the value of hyper-parameters to satisfy,

$$\sigma_w^2 \int \mathcal{D}_z [\phi'(\sqrt{q^*}z)]^2 = 1 \tag{18}$$

where $q^*$ is the fixed point of diagonal elements in the covariance matrix, and $\int \mathcal{D}z = \frac{1}{\sqrt{2\pi}} \int dz e^{-\frac{1}{2}z^2}$ is the measure for a normal distribution. For the ReLU activation, equation (18) requires $\sigma_w^2 = 2$ and $\sigma_b^2 = 0$.

The key idea is to study the asymptotic behavior of the normalized correlation defined as,

$$C_r(u, u') \equiv \frac{\Sigma_{(r)}(u, u')}{\sqrt{\Sigma_{(r)}(u, u)\, \Sigma_{(r)}(u', u')}}. \tag{19}$$

**Lemma 2** (Proposition 1 and 3 in Hayou et al. (2019a)). *Assume a network without aggregation, i.e., $L = 0$, with a Lipschitz nonlinearity $\phi$, then,*

- $\phi(x) = (x)_+$, $1 - C_r(u, u') \sim \frac{9\pi^2}{2r^2}$ *as* $r \to \infty$

- $\phi(x) = \tanh(x)$, $1 - C_r(u, u') \sim \frac{\beta}{r}$ *as* $r \to \infty$ *where* $\beta = \frac{2\int_{\mathcal{D}_z}[\phi'(\sqrt{q^*}z)^2]}{q\int_{\mathcal{D}_z}[\phi''(\sqrt{q^*}z)^2]}$.

*Proof.* We first decompose the integral calculation in the Equation (16) into two parts, diagonal elements and non-diagonal elements:

$$\Sigma_{(r)}(u, u) = \sigma_w^2 \int_{\mathcal{D}z} \phi^2\left(\sqrt{\Sigma_{(r-1)}(u, u)}z\right) + \sigma_b^2$$

$$\Sigma_{(r)}(u, u') = \sigma_w^2 \int_{\mathcal{D}z_1 \mathcal{D}z_2} \phi(u_1)\phi(u_2) + \sigma_b^2$$

where $u_1 = \sqrt{\Sigma_{(r-1)}(u, u)} z_1$, and $u_2 = \sqrt{\Sigma_{(r-1)}(u', u')}\left(C_{r-1}(u, u')z_1 + \sqrt{1 - C_{r-1}^2(u, u')}z_2\right)$, with $C_{r-1}(u, u') = \Sigma_{(r-1)}(u, u')/\sqrt{\Sigma_{(r-1)}(u, u)\Sigma_{(r-1)}(u', u')}$.

For simplicity, we define $q_r(u) = \Sigma_{(r)}(u, u)$, $q_r(u') = \Sigma_{(r)}(u', u')$. We then proceed with the proof by dividing the activation $\phi(x)$ into two classes, namely, ReLU and Tanh.

**ReLU activation, $\phi(x) = \max\{0, x\}$.** The recursive equation (16) for $q_r(u)$ reduces to,

$$q_r(u) = \frac{\sigma_w^2}{2}q_{r-1}(u) + \sigma_b^2$$

The edge of chaos condition $\sigma_w^2 \int \mathcal{D}_z[\phi'(\sqrt{q^*}z)]^2 = 1$ requires $\sigma_w^2 = 2$ and $\sigma_b^2 = 0$ for ReLU activation, which leads to,

$$\lim_{r \to \infty} q_r(u) = q_0(u) \equiv q(u)$$

Then the second integration for $C_r(u, u')$ becomes,

$$C_r(u, u') = \frac{\sigma_w^2 \int_{\mathcal{D}z_1 \mathcal{D}z_2} \phi\big(\sqrt{q_{r-1}(u)}z_1\big)\phi\big(\sqrt{q_{r-1}(u)}(C_{r-1}(u, u')z_1 + \sqrt{1 - C_{r-1}(u, u')^2}z_2)\big) + \sigma_b^2}{q_{r-1}(u)}$$

To investigate the propagation of $C_r(u, u')$, we set $q_r(u) = q_r(u') = q$, and define,

$$f(x) = \frac{\sigma_w^2 \int_{\mathcal{D}z_1 \mathcal{D}z_2} \phi\big(\sqrt{q}z_1\big)\phi\big(\sqrt{q}(xz_1 + \sqrt{1 - x^2}z_2)\big) + \sigma_b^2}{q}$$

Let $x \in [0, 1]$, the derivative of $f(x)$ satisfies,

$$f'(x) = 2\int_{\mathcal{D}z_1 \mathcal{D}z_2} 1_{z_1 > 0} 1_{xz_1 + \sqrt{1 - x^2}z_2 > 0}$$

This can be seen from a simple derivation,

$$f'(x) = \int_{\mathcal{D}z_1}\int_{\mathcal{D}z_2} \phi(z_1)\phi'(xz_1 + \sqrt{1 - x^2}z_2)(z_1 - \frac{x}{\sqrt{1 - x^2}}z_2)$$

$$= \int_{\mathcal{D}z_1}\int_{\mathcal{D}z_2} \phi(z_1)\phi'(xz_1 + \sqrt{1 - x^2}z_2)(z_1) - \int_{\mathcal{D}z_1}\int_{\mathcal{D}z_2} \phi(z_1)\phi'(xz_1 + \sqrt{1 - x^2}z_2)(\frac{x}{\sqrt{1 - x^2}}z_2)$$

Using an identity for Gaussian integration $\int_{\mathcal{D}z} zg(z) = \int_{\mathcal{D}z} g'(z)$ yields,

$$f'(x) = \int_{\mathcal{D}z_1}\int_{\mathcal{D}z_2} \big[\phi'(z_1)\phi'(xz_1 + \sqrt{1 - x^2}z_2) + \phi(z_1)\phi''(xz_1 + \sqrt{1 - x^2}z_2) - \phi(z_1)\phi''(xz_1 + \sqrt{1 - x^2}z_2)\big]$$

$$= \int_{\mathcal{D}z_1}\int_{\mathcal{D}z_2} \phi'(z_1)\phi'(xz_1 + \sqrt{1 - x^2}z_2)$$

Then the second derivative of $f(x)$ becomes, $f''(x) = \frac{1}{\pi\sqrt{1 - x^2}}$. So using the equation above and the condition $f'(0) = \frac{1}{2}$, we can obtain another form of the derivative of $f(x)$,

$$f'(x) = \frac{1}{\pi}\arcsin(x) + \frac{1}{2}$$

Because $\int \arcsin = x\arcsin + \sqrt{1 - x^2}$ and $f(1) = 1$, then for $x \in [0, 1]$,

$$f(x) = \frac{2x\arcsin(x) + 2\sqrt{1 - x^2} + x\pi}{2\pi}$$

Substituting $f(x) = C_r(u, u')$ and $x = C_{r-1}(u, u')$, into expression above here, we have,

$$C_r(u, u') = \frac{2C_{r-1}(u, u')\arcsin(C_{r-1}(u, u')) + 2\sqrt{1 - C_{r-1}^2(u, u')} + C_{r-1}(u, u')\pi}{2\pi}$$

Now we study the behavior of $C_r(u, u')$ as $r$ tends to infinity. Using Taylor expansion, we have,

$$f(x)|_{x \to 1-} = x + \frac{2\sqrt{2}}{3\pi}(1 - x)^{3/2} + O((1 - x)^{5/2})$$

By induction analysis, the sequence $C_r(u, u')$ increases to the fixed point 1. Besides, we can replace $x$ with $C_r(u, u')$,

$$C_{r+1}(u, u') = C_r(u, u') + \frac{2\sqrt{2}}{3\pi}(1 - C_r(u, u'))^{3/2} + O((1 - C_r(u, u'))^{5/2})$$

Let $\gamma_r = 1 - C_r(u, u')$, then we have,

$$\gamma_{r+1} = \gamma_r - \frac{2\sqrt{2}}{3\pi}\gamma_r^{3/2} + O(\gamma_r^{5/2})$$

so that,

$$\gamma_{r+1}^{-1/2} = \gamma_r^{-1/2}(1 - \frac{2\sqrt{2}}{3\pi}\gamma_r^{1/2} + O(\gamma_r^{3/2}))^{-1/2} = \gamma_r^{-1/2} + \frac{\sqrt{2}}{3\pi} + O(\gamma_r)$$

As $r$ tends to infinity, we have,

$$\gamma_{r+1}^{-1/2} - \gamma_r^{-1/2} \sim \frac{\sqrt{2}}{3\pi}$$

It means,

$$\gamma_r^{-1/2} \sim \frac{\sqrt{2}}{3\pi}r$$

Therefore, we have,

$$1 - C_r(u, u') \sim \frac{9\pi^2}{2r^2}$$

**Tanh activation,** $\phi(x) = \tanh(x)$. For the argument of fixed point, we ask readers to refer to Hayou et al. (2019a). Here, we only discuss the convergence rate of GPK, which means we directly assume that $f(x)$ tends to 1 as the depth tends to infinity $\lim_{r \to \infty} C_r(u, u') = 1$. For the function $f(x)$, a Taylor expansion near 1 yields,

$$f(x) = 1 + (x - 1)f'(1) + \frac{(x-1)^2}{2}f''(1) + O((x-1)^{5/2})$$

where $f'(1) = \sigma_w^2 \int_{\mathcal{D}_z}[\phi'(\sqrt{q}z)^2]$, and $f''(1) = \sigma_w^2 q \int_{\mathcal{D}_z}[\phi''(\sqrt{q}z)^2]$. Denote $\gamma_r = 1 - C_r(u, u')$, then we have,

$$\gamma_{r+1} = \gamma_r - \frac{\gamma_r^2}{\beta} + O(\gamma_l^{5/2})$$

where $\beta = \frac{2\int_{\mathcal{D}_z}[\phi'(\sqrt{q^*}z)^2]}{q\int_{\mathcal{D}_z}[\phi''(\sqrt{q^*}z)^2]}$. Therefore,

$$\gamma_{r+1}^{-1} = \gamma_r^{-1}(1 - \frac{\gamma_r}{\beta} + O(\gamma_r^{3/2})) = \gamma_r^{-1} + \frac{1}{\beta} + O(\gamma_r^{1/2}).$$

Thus, we have,

$$1 - C_{(r)}(u, u') \sim \frac{\beta}{r} \text{ as } r \to \infty$$

$\square$

Lemma 2 shows that the covariance matrix converges to a constant matrix at a polynomial rate of $1/r^2$ for ReLU and of $1/r$ for tanh activation on the edge of chaos. This implies that a network without aggregation could retain its expressivity at a large depth. However, for general cases, due to aggregation, the rate of convergence would change from polynomial to exponential. This would cause the trainability of deep GNNs to be problematic, as shown in the following section.

### B.3 CONVERGENCE OF GPK FOR GCNS

Then we formally characterize the convergence of Gaussian Process kernel $\Sigma_{(r)}^{(l)}(u, u')$ of a GCN in the infinite-width limit:

**Lemma 3.** *If $\mathcal{A}(G)$ is ireducible and aperiodic, with a stationary distribution vector $\vec{\pi}(G)$, then there exist constants $0 < \alpha < 1$ and $C > 0$, and constant vector $\vec{v} \in \mathbb{R}^{n^2 \times 1}$ depending on the number of MLP iterations $R$, such that*

$$|\Sigma_{(r)}^{(l)}(u, u') - \vec{\pi}(G)^T\vec{v}| \leq C\alpha^l$$

*Proof.* We prove the result via an induction method. For $l = 0$, according to the Cauchy-Buniakowsky-Schwarz Inequality

$$\Sigma_{(0)}^{(0)}(u, u') = h_u^{(0)}h_{u'}^{(0)} \leq \|h_u^{(0)}\|_2\|h_{u'}^{(0)}\|_2 = 1$$

Thus, there is a constant $C$, depending on $G(V, E)$, and the number of MLP operations $R$, over feature initialization, such that,

$$\left| \Sigma^{(0)}_{(0)}(u, u') - \vec{\pi}(G)^T \vec{v} \right| < C$$

Assume the result is valid for $\Sigma^{(l)}_{(r)}(u, u')$, then we have,

$$\left| \Sigma^{(l)}_{(r)}(u, u') - \vec{\pi}(G)^T \vec{v} \right| \leq C_0 \alpha^l$$

where $C_0$ is a constant satisfying $0 < C_0 < C$. Now we consider the distance between $\Sigma^{(l+1)}_{(r)}(u, u')$ and $C\alpha^{l+1}$. To compute this, we need to divide the propagation from $l$ layer to $l+1$ layer into three steps:

1. $\Sigma^{(l)}_{(r)} \to \Sigma^{(l)}_{(r+1)} \to \cdots \to \Sigma^{(l)}_{(R)}$

2. $\Sigma^{(l)}_{(R)} \to \Sigma^{(l+1)}_{(0)}$

3. $\Sigma^{(l+1)}_{(0)} \to \Sigma^{(l+1)}_{(1)} \to \cdots \to \Sigma^{(l+1)}_{(r)}$.

It is not hard to find that steps 1 and 3 correspond to non-linear transformation while step 2 corresponds to aggregation operation, we then characterize them one by one,

**MLP Propagation** The assumption $\left| \Sigma^{(l)}_{(r)}(u, u') - \vec{\pi}(G)^T \vec{v} \right| \leq C_0 \alpha^l$ implicitly implies that $C_r(u, u')$ is close to one. Because $C_r(u, u') = \frac{\Sigma^{(l)}_{(r)}(u, u')}{\sqrt{\Sigma^{(l)}_{(r)}(u, u)\, \Sigma^{(l)}_{(r)}(u', u')}}$, then,

$$\frac{\vec{\pi}(G)^T \vec{v} - C_0 \alpha^l}{\vec{\pi}(G)^T \vec{v} + C_0 \alpha^l} \leq C_r(u, u') \leq \frac{\vec{\pi}(G)^T \vec{v} + C_0 \alpha^l}{\vec{\pi}(G)^T \vec{v} - C_0 \alpha^l}$$

Because $C_0 \alpha^l \ll \vec{\pi}(G)^T \vec{v}$, we have $C_r(u, u') = 1 + O(\alpha^l)$. Recall the property of MLP propagation function $f(x)$ for $x = C_r(u, u')$, when $C_r(u, u')$ is close to 1:

$$f(x)|_{x \to 1-} = x + \frac{2\sqrt{2}}{3\pi}(1-x)^{3/2} + O((1-x)^{5/2})$$

This implies,

$$|C_{(r+1)}(u, u') - C_{(r)}(u, u')| = \frac{2\sqrt{2}}{3\pi}(1 - C_{(r)}(u, u'))^{\frac{3}{2}} + O(1 - C_{(r)}(u, u'))^{\frac{5}{2}} = O(\alpha^{\frac{3}{2}l})$$

With this result, we can further obtain,

$$
\begin{aligned}
|\Sigma^{(l)}_{(r+1)}(u, u') - \vec{\pi}(G)^T \vec{v}| &= |\Sigma^{(l)}_{(r+1)}(u, u') - \Sigma^{(l)}_{(r)}(u, u') + \Sigma^{(l)}_{(r)}(u, u') - \vec{\pi}(G)^T \vec{v}| \\
&\leq |\Sigma^{(l)}_{(r+1)}(u, u') - \Sigma^{(l)}_{(r)}(u, u')| + |\Sigma^{(l)}_{(r)}(u, u') - \vec{\pi}(G)^T \vec{v}| \\
&= |C^{(l)}_{(r+1)}(u, u') - C^{(l)}_{(r)}(u, u')|\sqrt{\Sigma^{(l)}_{(r)}(u, u)\Sigma^{(l)}_{(r)}(u', u')} + |\Sigma^{(l)}_{(r)}(u, u') - \vec{\pi}(G)^T \vec{v}| \\
&= C_1 \alpha^{\frac{3}{2}l} + C_0 \alpha^l \leq (C_0 + C_1)\alpha^l.
\end{aligned}
$$

where $C_1$ is a positive constant. Repeat the proof process, we have a relation for $\Sigma^{(l)}_{(R)}(u, u')$ at the last step of 1,

$$|\Sigma^{(l)}_{(R)}(u, u') - \vec{\pi}(G)^T \vec{v}| \leq (C_0 + (R-r)C_1)\alpha^l. \tag{20}$$

**Aggregation Propagation**   We go through an aggregation operation $\mathcal{A}(G)$. In this case, we use a matrix form, and take equation (20) into aggregation function, yielding the result as follows,

$$\vec{\Sigma}^{(l+1)}_{(0)}(G) = \mathcal{A}(G)\vec{\Sigma}^{(l)}_{(R)}(G) = \mathcal{A}(G)(\Pi(G)\vec{v} + \vec{O}(\alpha^l))$$

where $\vec{O}(\alpha^l) \in \mathbb{R}^{n^2 \times 1}$ denotes a vector in which every element is bounded by $\alpha^l$.

According to Theorem 4.9 in Freedman (2017), we have $\mathcal{A}^l \Pi(G)\vec{v} = \Pi(G)\vec{v}$ as $l \to \infty$. When $l$ is finite, the error is of exponential decay. Here, we use $0 < \alpha < 1$ to denote the corresponding base number. Therefore,

$$|\Sigma^{(l+1)}_{(0)}(u, u') - \vec{\pi}(G)^T \vec{v}| \le (C_0 + (R-r)C_1)\alpha^{l+1}.$$

Finally, by repeating the result in step 1, we have,

$$|\Sigma^{(l+1)}_{(r)}(u, u') - \vec{\pi}(G)^T \vec{v}| \le (C_0 + RC_1)\alpha^{l+1} = C\alpha^{l+1}.$$

$$\square$$

**Remark 2.** *In the proof, there is a $RC_1$ term in each propagation of $l \to l+1$, which may lead to explosion when $l$ tends to infinity. Basically, this problem can be solved by a careful analysis, because the constant $C_1$ is associated with $O(\alpha^{\frac{3}{2}l})$, which has a smaller order compared to $O(\alpha^l)$.*

### B.4   CONVERGENCE OF GNTK

Finally, we arrive at our main theorem:

**Theorem 1** (Convergence rate of GNTK). *If transition matrix $\mathcal{A}(G) \in \mathbb{R}^{n^2 \times n^2}$ is irreducible and aperiodic, with a stationary distribution vector $\vec{\pi}(G) \in \mathbb{R}^{n^2 \times 1}$, where $\vec{\Theta}^{(l)}_{(0)}(G) = \mathcal{A}(G)\vec{\Theta}^{(l-1)}_{(R)}(G)$ and $\vec{\Theta}^{(l)}_{(r)}(G) \in \mathbb{R}^{n^2 \times 1}$ is the result of being vectorized. Then there exist constants $0 < \alpha < 1$ and $C > 0$, and constant vectors $\vec{v}, \vec{v}' \in \mathbb{R}^{n^2 \times 1}$ depending on the number of MLP iterations $R$, such that $\left|\Theta^{(l)}_{(r)}(u, u') - \vec{\pi}(G)^T\left(Rl\vec{v} + \vec{v}'\right)\right| \le C\alpha^l$.*

*Proof.* This proof has the same strategy to that of Lemma 3. The first step is to understand Equation (9) in the large-depth limit.

$$\Theta^{(l)}_{(r)}(u, u') = \Theta^{(l)}_{(r-1)}(u, u')\dot{\Sigma}^{(l)}_{(r)}(u, u') + \Sigma^{(l)}_{(r)}(u, u')$$

According to the result of Lemma 3, we have already known,

$$\Sigma^{(l)}_{(r)}(u, u') = \vec{\pi}(G)^T\vec{v} + O(\alpha^l)$$

To proceed the proof, we need to work out the behavior of $\dot{\Sigma}^{(l)}_{(r)}(u, u')$ in the large depth.

**ReLU activation, $\phi(x) = \max\{0, x\}$.**   Recall that we define $C_{r+1} = f(C_r)$, and we have,

$$f'(x) = \frac{1}{\pi}\arcsin(x) + \frac{1}{2}$$

Then, at the edge of chaos,

$$\dot{\Sigma}_{(r)}(u, u') = f'(C_r(u, u')) = \frac{1}{\pi}\arcsin(C_r(u, u')) + \frac{1}{2}$$
$$= 1 - \frac{2}{\pi}(1 - C_r(u, u'))^{1/2} + O(1 - C_r(u, u'))^{3/2} = 1 + O(\alpha^{l/2})$$

**Tanh activation, $\phi(x) = \tanh(x)$.** We have

$$f'(x) = 1 - (x-1)f''(1) + O((x-1)^2)$$

At the edge of chaos,

$$\dot{\Sigma}_{(r)}(u, u') = 1 + O(\alpha^l)$$

Now we prove the result via an induction method. For $l = 0$, we directly have,

$$\Theta_{(0)}^{(0)}(u, u') = \Sigma_{(0)}^{(0)}(u, u') \le \|h_u^{(0)}\|_2 \|h_{u'}^{(0)}\|_2 = 1$$

Thus there is a constant $C$, depending on $G(V, E)$, and the number of MLP operations $R$, over feature initialization,

$$|\Theta_{(0)}^{(0)}(u, u') - \vec{\pi}(G)^T \vec{v}'| < C$$

Assume the result is valid for $\Theta_{(r)}^{(l)}(u, u')$, then we have,

$$|\Theta_{(r)}^{(l)}(u, u') - \vec{\pi}(G)^T (Rl\vec{v} + \vec{v}')| \le C\alpha^l$$

Now we consider the distance between $\Theta_{(r)}^{(l+1)}(u, u')$ and $\vec{\pi}(G)^T(R(l+1)\vec{v} + \vec{v}'')$. To prove this, we need to divide the propagation from $l$ layer to $l+1$ layer into three steps:

1. $\Theta_{(r)}^{(l)} \to \Theta_{(r+1)}^{(l)} \to \cdots \to \Theta_{(R)}^{(l)}$

2. $\Theta_{(R)}^{(l)} \to \Theta_{(0)}^{(l+1)}$

3. $\Theta_{(0)}^{(l+1)} \to \Theta_{(1)}^{(l+1)} \to \cdots \to \Theta_{(r)}^{(l+1)}$.

For step 1, we have,

$$|\Theta_{(r+1)}^{(l)}(u, u') - \vec{\pi}(G)^T((lR+1)\vec{v} + \vec{v}')|$$
$$= |\Sigma_{(r+1)}^{(l)}(u, u') + \dot{\Sigma}_{(r+1)}^{(l)}(u, u')\Theta_{(r)}^{(l)}(u, u') - \vec{\pi}(G)^T((lR+1)\vec{v} + \vec{v}')|$$
$$= |\vec{\pi}(G)^T\vec{v}(1 + O(\alpha^l)) + (1 + O(\alpha^{l/2}))\Theta_{(r)}^{(l)}(u, u') - \vec{\pi}(G)^T((lR+1)\vec{v} + \vec{v}')| \le C\alpha^l$$

Repeat the process, we have a relation for $\Theta_{(R)}^{(l)}(u, u')$ at the last transformation in step 1,

$$|\Theta_{(R)}^{(l)}(u, u') - \vec{\pi}(G)^T((lR+R-r)\vec{v} + \vec{v}')| \le C\alpha^l.$$

Secondly, we go through an aggregation operation. Because it is a Markov chain step, we have,

$$|\Theta_{(0)}^{(l+1)}(u, u') - \vec{\pi}(G)^T((Rl+R-r)\vec{v} + \vec{v}')| \le C\alpha^{l+1}.$$

Repeat the result in step 1, we finally have,

$$|\Theta_{(r)}^{(l+1)}(u, u') - \vec{\pi}(G)^T((Rl+R)\vec{v} + \vec{v}')|$$
$$= |\Theta_{(r)}^{(l+1)}(u, u') - \vec{\pi}(G)^T(R(l+1)\vec{v} + \vec{v}'| \le C\alpha^{l+1}.$$

$\square$

## C    Trainability of Ultra-Wide GCNs

**Corollary 1** (Trainability of ultra-wide GCNs). *Consider a GCN of the form (6) and (7), with depth $l$, number of non-linear transformations $r$, an MSE loss, and a Lipchitz activation, trained with gradient descent on a node classification task. Then, the output function follows, $f_t(X) = e^{-\eta\Theta_{(r)}^{(l)}(G)t}f_0(X) + (I - e^{-\eta\Theta_{(r)}^{(l)}(G)t}Y)$, where $X$ and $Y$ are node features and labels from training set. Then, $\Theta_{(r)}^{(l)}(G)$ is singular when $l \to \infty$. Moreover, there exists a constant $C > 0$ such that for all $t > 0$, $\|f_t(X) - Y\| > C$.*

*Proof.* According to the result from Jacot et al. (2018), GNTK $\Theta_{(r)}^{(l)}(G)$ converges to a deterministic kernel and remains constant during gradient descent in the infinite-width limit. We omit the proof procedure for this result, since it is now a standard conclusion in the NTK study.

Based on the conclusion above, Lee et al. (2019a) proved that the infinitely-wide neural network is equivalent to its linearized mode,

$$f_t^{\text{lin}}(X) = f_0(X) + \nabla_\theta f_0(X)|_{\theta=\theta_0}\omega_t$$

where $\omega_t = \theta_t - \theta_0$. We call it a linearized model because only zero and first order term of Taylor expansion are kept. Since we know the dynamics of gradient flow using this linearized function are governed by,

$$\dot{\omega}_t = -\eta\nabla_\theta f_0(X)^T\nabla_{f_t^{\text{lin}}(X)}\mathcal{L}$$

$$\dot{f}_t^{\text{lin}}(X) = -\eta\Theta_{(r)}^{(l)}(G)\nabla_{f_t^{\text{lin}}(X)}\mathcal{L}$$

where $\mathcal{L}$ is an MSE loss, then the above equations have closed form solutions

$$f_t^{\text{lin}}(X) = e^{-\eta\Theta_{(r)}^{(l)}(G)t}f_0(X) + (I - e^{-\eta\Theta_{(r)}^{(l)}(G)t})Y$$

Since Lee et al. Lee et al. (2019a) showed that $f_t^{\text{lin}}(X) = f_t(X)$ in the infinite width limit, thus we have,

$$f_t(X) = e^{-\eta\Theta_{(r)}^{(l)}(G)t}f_0(X) + (I - e^{-\eta\Theta_{(r)}^{(l)}(G)t})Y \tag{21}$$

In Theorem 1, we have shown $\Theta_{(r)}^{(l)}(G)$ converges to a constant matrix at an exponential rate, thus being singular in the large depth limit. According to Equation (21), we know that,

$$\|f_t(X) - Y\| = \|e^{-\eta\Theta_{(r)}^{(l)}(G)t}(f_0(X) - Y\|$$

Let $\Theta_{(r)}^{(\infty)}$ be the GNTK in the large depth limit, $\Theta_{(r)}^{(\infty)}(G) = Q^TDQ$ be the decomposition of the GNTK, where $Q$ is an orthogonal matrix and $D$ is a diagonal matrix. Because $\Theta_{(r)}^{(\infty)}(G)$ is singular, $D$ has at least one zero value $d_j = 0$, then

$$\|f_t(X) - Y\| = \|Q^T(f_t(X) - Y)Q\| \geq \|[Q^T(f_0(X) - Y)Q]_j\|$$

□

**Remark 3.** *In the proof we assume the loss is MSE. Nevertheless, the conclusion regarding trainability can be applied to other common losses such as cross-entropy. For cross-entropy loss, even though we cannot derive an analytical expression for the solution, we can prove that the GNTK still governs the trainability and the law of GNTK is not affected by the loss. Thus, the results for trainability still hold in the cross-entropy case.*

## D CONVERGENCE OF RESIDUAL GNTK

### D.1 RESIDUAL CONNECTION IN AGGREGATION

**Theorem 2** (Convergence rate of residual connection in aggregation). *Consider a GNTK of nonlinear transformation (9) and residual connection (12). Then with a stationary vector $\tilde{\tilde{\pi}}(G)$ for $\tilde{\mathcal{A}}(G)$, there exist constants $0 < \tilde{\alpha} < 1$ and $C > 0$, and constant vectors $\vec{v}$ and $\vec{v}'$ depending on $R$, such that $\left|\Theta_{(r)}^{(l)}(u, u') - \tilde{\tilde{\pi}}(G)^T(Rl\vec{v} + \vec{v}')\right| \leq C\tilde{\alpha}^l$. Furthermore, we denote the second largest eigenvalue of $\mathcal{A}(G)$ and $\tilde{\mathcal{A}}(G)$ as $\lambda_2$ and $\tilde{\lambda}_2$, respectively. Then, $\tilde{\lambda}_2 > \lambda_2$.*

*Proof.* According to the aggregation function for covariance matrix, we have

$$\vec{\Theta}^{(l)}(G) = (1 - \delta)\mathcal{A}(G)\vec{\Theta}^{(l-1)}(G) + \delta\vec{\Theta}^{(l-1)}(G)$$
$$= ((1 - \delta)\mathcal{A} + \delta I)\vec{\Theta}^{(l-1)}(G) = \tilde{\mathcal{A}}(G)\vec{\Theta}^{(l-1)}(G)$$

The new aggregation matrix $\tilde{\mathcal{A}}(G)$ is a stochastic transition matrix as well, which can be seen from,

$$\sum_j \tilde{\mathcal{A}}(G)_{ij} = (1-\delta)\sum_j \mathcal{A}(G)_{ij} + \delta\sum_j I_{ij} = 1$$

Then we compare the second largest eigenvalue between two transition matrices. Suppose the eigenvalues of the original matrix $\mathcal{A}(G)$ are $\{\lambda_1 > \lambda_2 > \cdots > \lambda_{n^2}\}$. We already know that the maximum eigenvalue is $\lambda_1 = 1$, and the converge rate is governed by the second largest eigenvalue $\lambda_2$. Now we consider the second largest eigenvalue $\tilde{\lambda}_2$ of matrix $\tilde{\mathcal{A}}$:

$$\tilde{\lambda}_2 = (1-\delta)\lambda_2 + \delta = \lambda_2 + \delta(1-\lambda_2) > \lambda_2$$

$\square$

**Remark 4.** *The theorem above gives us a certain conclusion about the relationship between $\lambda_2$ and $\tilde{\lambda}_2$. Here, we discuss more about the relationship between $\alpha$ and $\tilde{\alpha}$. According to (Rosenthal, 1995), the relationship between convergence rate $\alpha$ and the second largest eigenvalue $\lambda_2$ of transition matrix $\mathcal{A}(G)$ can be expressed as,*

$$\left|\Theta^{(l)}_{(r)}(u,u') - \vec{\pi}(G)^T\left(Rl\vec{v} + \vec{v}'\right)\right| \le Cl^{J-1}\lambda_2^{l-J} \le C\alpha^l$$

*where $\alpha > \lambda_2$, and $J > 1$ is the size of the largest Jordan block of $\mathcal{A}(G)$. From the above inequality, we can conclude that the second largest eigenvalue almost governs the convergence rate. However, to maintain rigor, we cannot directly obtain $\alpha < \tilde{\alpha}$. Instead, from the result that $\tilde{\lambda}_2 > \lambda_2$, we say that $\tilde{\alpha}$ is roughly larger than $\alpha$.*

### D.2 RESIDUAL CONNECTION IN MLP

**Theorem 3** (Convergence rate of GNTK with residual connection between transformations)**.** *Consider a GNTK of the form (8) and (12). If $\mathcal{A}(G)$ is irreducible and aperiodic, with a stationary distribution $\pi(G)$, then there exist constants $0 < \alpha < 1$ and $C > 0$, and constant vectors $v$ and $v'$ depending on $R$, such that, $|\Theta^{(l)}_{(r)}(u,u') - \vec{\pi}(G)^T \cdot \left(Rl(1 + \frac{\sigma_w^2}{2})^{Rl}\vec{v} + \vec{v}'\right)| \le C\alpha^l$.*

*Proof.* For residual connection in MLP, we have,

$$q_r(u) = q_{r-1}(u) + \frac{\sigma_w^2}{2}q_{r-1}(u) = (1 + \frac{\sigma_w^2}{2})q_{r-1}(u)$$

Since $\sigma_w^2 > 0$, $q_r(u)$ grows at an exponential rate.

Now, we turn to compute the correlation term $C_r(u,u')$. For convenience, we suppose $q_r(u) = q_r(u')$. Then,

$$C_{r+1}(u,u') = \frac{\Sigma_{r+1}(u,u')}{q_{r+1}(u)} = \frac{1}{1 + \frac{\sigma_w^2}{2}}\frac{\Sigma_r(u,u')}{q_r(u)} + \frac{1}{1 + \frac{\sigma_w^2}{2}}\frac{\sigma_w^2}{2}f(C_r(u,u'))$$

$$= \frac{1}{1 + \frac{\sigma_w^2}{2}}C_r(u,u') + \frac{\frac{\sigma_w^2}{2}}{1 + \frac{\sigma_w^2}{2}}f(C_r(u,u'))$$

Using Taylor expansion of $f$ near 1, as have been done in the proof of Lemma 2,

$$f(x)|_{x\to 1-} = x + \frac{2\sqrt{2}}{3\pi}(1-x)^{3/2} + O((1-x)^{5/2})$$

we have,

$$C_{r+1}(u,u') = C_r(u,u') + \frac{2\sqrt{2}}{3\pi}\frac{\frac{\sigma_w^2}{2}}{1 + \frac{\sigma_w^2}{2}}\left[(1 - C_r(u,u')^{3/2} + O((1 - C_r(u,u')^{5/2}))\right]$$

Note that it is similar to the case of MLP without residual connection:

$$C_{r+1}(u, u') = C_r(u, u') + \frac{2\sqrt{2}}{3\pi}\big[(1 - C_r(u, u')^{3/2} + O((1 - C_r(u, u')^{5/2}))\big]$$

Following the proof diagram in Lemma 3 and Theorem 1, we can obtain the behavior of $\Sigma^{(l)}_{(r)}(u, u')$ and $\Theta^{(l)}_{(r)}(u, u')$ in the large depth limit,

$$\Big|\Sigma^{(l)}_{(r)}(u, u') - \vec{\pi}(G)^T\big((1 + \frac{\sigma_w^2}{2})^{Rl}\vec{v}\big)\Big| \le C\alpha^l$$

$$\Big|\Theta^{(l)}_{(r)}(u, u') - \vec{\pi}(G)^T\big(Rl(1 + \frac{\sigma_w^2}{2})^{Rl}\vec{v} + \vec{v}'\big)\Big| \le C\alpha^l$$

$\square$

# E    APPENDIX: DISCUSSION ON THE TRAINABILITY OF GNTK WITH CRTICAL DROPEDGE

According to the critical percolation theory, there exists a large connected component of order $O(n)$ (Erdős & Rényi, 1961; Erdös & Rényi, 2011), where $n$ is the number of nodes in a graph. This implies that there exists a (connected) graph of size $O(n)$ when we use the critical dropping rate. To preserve information, edges would be resampled at every iteration. This implies that at each iteration during training, the GNN takes a random and large graph. For the whole training process, we can think that we have used all the information conveyed by the graph.

It would be interesting to consider deriving some theoretical results for the limiting behavior of the GNTK in the large depth with Critical DropEdge. Since Critical DropEdge is originated from the random graph theory, a promising approach is to analyze spectral distributions of adjacency matrix and Laplacian matrix with a random graph model, like existing studies (Ding & Jiang, 2010; Chung & Radcliffe, 2011).

# F    APPENDIX: DATASETS AND IMPLEMENTATION DETAILS

## F.1    DATASETS

For node classification tasks, Cora, Citeseer and Pubmed (Kipf & Welling, 2017)) are three commonly used citation networks, and Physics (Shchur et al., 2018) is a co-author network. Detailed information of the four benchmark datasets is listed as follows and summarized in Table 3.

- The Cora dataset consists of 2,708 scientific publications classified into one of seven classes, and 5,429 links. Each publication is described by a 0/1-valued word vector indicating the absence/presence of the corresponding word from the dictionary. The dictionary consists of 1,433 unique words.

- The Citeseer dataset consists of 3,312 scientific publications classified into one of six classes, and 4,732 links. Each publication is described by a 0/1-valued word vector indicating the absence/presence of the corresponding word from the dictionary. The dictionary consists of 3,703 unique words.

- The Pubmed Diabetes dataset consists of 19,717 scientific publications from PubMed database pertaining to diabetes classified into one of three classes. The citation network consists of 44,338 links. Each publication is described by a TF-IDF weighted word vector from a dictionary comprised of 500 unique words.

- The Physics dataset consists of 34,493 authors as nodes, and edges indicate whether two authors have co-authored a paper. Node features are paper keywords from the author's papers. Following Kim & Oh (2020), we reduce the original dimension (6,805 and 8,415) to 500 using PCA. The split is the 20-per-class/30-per-class/rest. The goal of this task is to classify each author's respective field of study.

Table 3: Details of Datasets

| Dataset | Nodes | Edges | Classes | Features | Critical Edge Sampling | Train/Val/Test |
|---|---|---|---|---|---|---|
| Cora | 2,708 | 5,429 | 7 | 1,433 | 24.94% | 0.05/0.18/0.37 |
| Citeseer | 3,327 | 4,732 | 6 | 3,703 | 35.15% | 0.04/0.15/0.30 |
| Pubmed | 19,717 | 44,338 | 3 | 500 | 22.23% | 0.003/0.03/0.05 |
| Physics | 34,493 | 247,962 | 5 | 500 | 6.96 % | 0.003/0.004/0.99 |

Table 4: Comparison results of test accuracy (%) with different infinite-wide backbones.

| Dataset | Backbone | 4-layer | | 8-layer | | 32-layer | | 64-layer | |
|---|---|---|---|---|---|---|---|---|---|
| | | Orignal | C-DropEdge | Original | C-DropEdge | Original | C-DropEdge | Original | C-DropEdge |
| Cora | GCN | 79.98 | **79.98** | 79.48 | **80.61** | 73.75 | **75.86** | 72.59 | **74.03** |
| | JKNet | 78.27 | 77.29 | 80.26 | 80.02 | 78.00 | **79.01** | 75.90 | **76.75** |
| Citeseer | GCN | 66.38 | **66.97** | 54.42 | **64.88** | 50.72 | **51.22** | 43.77 | **46.49** |
| | JKNet | 67.04 | **67.67** | 65.54 | **66.45** | 54.57 | **56.39** | 46.71 | **47.96** |
| Pubmed | GCN | 74.59 | **74.77** | 73.58 | **76.67** | 71.63 | **74.77** | 66.28 | **71.72** |
| | JKNet | 75.30 | 74.35 | 76.42 | **77.17** | 75.71 | **76.68** | 74.81 | **75.74** |

## F.2 EXPERIMENTAL IMPLEMENTATION

We use the PyTorch implementation to simulate both infinitely-wide and finitely-wide GCNs:

- The **infinitely-wide** GCNs use part of code from Du et al. (2019b), which is originally adopted for graph classification. We redesign the calculation method of GNTK (Du et al., 2019b)[2] according to the formula in Section 3.1 and use it to process node classification tasks.

- For **finitely-wide** GNNs with DropEdge (Rong et al., 2019)[3], we use the code from (Rong et al., 2019), we perform random hyper-parameter search for each model, and report the case giving the best accuracy on validation set of each benchmark, following the same strategy as Rong et al. (2019). The difference is that, in each experiment, we fix the edge sampling percentage as $\rho = \frac{|V|}{2|E|}$, which listed in the last column of Table 3.

All codes mentioned above use the MIT license. All experiments are conducted on two Nvidia Quadro RTX 6000 GPUs.

## G APPENDIX: ADDITIONAL EXPERIMENTS AND DISCUSSION

### G.1 INFINITELY-WIDE BACKBONES

In the main paper, we reported the performance of GCNs with finite-width, here we show more results with respect to infinitely-wide GCNs, as shown in Table 4. We apply Critical DropEdge to infinitely-wide backbones on semi-supervised node classification. We consider two backbones, which are GCN (Kipf & Welling, 2017) and JKNet (Xu et al., 2018), and the edge preserving percentage $\rho(G)$ is shown in Table 3. First, we calculate a GNTK with respect to the graph data and GNN. Then, we apply kernel regression with calculated GNTK to complete node classification task.

Table 4 summaries the performance on three citation networks. In this table, we report the performance of GCNs with 4/8/32/64 layers. It is shown that, in most cases, using Critical DropEdge (C-DropEdge) can achieve better results than original GNNs in the infinitely-wide case, especially when the model is deep.

---

[2]We use the implementation of GNTK available at `https://github.com/KangchengHou/gntk`.
[3]We use the DropEdgeimplementation available at `https://github.com/DropEdge/DropEdge`.

Table 5: Comparison results of test accuracy (%) between C-DropEdge and DropEdge.

| Dataset | Backbone Finite | 4-layer Original | DropEdge | C-DropEdge | 8-layer Original | DropEdge | C-DropEdge |
|---|---|---|---|---|---|---|---|
| Cora | GCN | $79.8 \pm 1.1$ | $80.4 \pm 1.4$ | $\mathbf{82.0 \pm 1.2}$ | $73.2 \pm 2.7$ | $75.1 \pm 2.4$ | $\mathbf{77.3 \pm 2.5}$ |
| | JKNet | $81.1 \pm 1.0$ | $82.2 \pm 0.7$ | $\mathbf{82.5 \pm 0.7}$ | $80.9 \pm 1.2$ | $82.0 \pm 0.9$ | $\mathbf{82.1 \pm 0.5}$ |
| | IncepGCN | $80.0 \pm 0.9$ | $80.6 \pm 1.2$ | $\mathbf{82.4 \pm 0.5}$ | $78.6 \pm 1.7$ | $81.2 \pm 1.3$ | $\mathbf{82.3 \pm 0.6}$ |
| Citeseer | GCN | $61.2 \pm 3.0$ | $63.7 \pm 2.5$ | $\mathbf{69.0 \pm 0.8}$ | $50.2 \pm 5.7$ | $52.8 \pm 5.1$ | $\mathbf{54.1 \pm 5.9}$ |
| | JKNet | $69.6 \pm 1.2$ | $70.2 \pm 1.0$ | $\mathbf{70.3 \pm 0.7}$ | $70.7 \pm 1.0$ | $70.5 \pm 1.1$ | $\mathbf{70.8 \pm 1.2}$ |
| | IncepGCN | $69.4 \pm 1.5$ | $70.0 \pm 1.0$ | $\mathbf{70.8 \pm 0.6}$ | $69.0 \pm 1.2$ | $70.8 \pm 1.1$ | $\mathbf{70.9 \pm 0.5}$ |
| Pubmed | GCN | $77.4 \pm 0.7$ | $77.6 \pm 1.4$ | $\mathbf{78.0 \pm 0.4}$ | $57.2 \pm 8.4$ | $57.5 \pm 6.1$ | $\mathbf{58.9 \pm 6.9}$ |
| | JKNet | $76.5 \pm 0.9$ | $77.1 \pm 1.1$ | $\mathbf{77.4 \pm 1.0}$ | $76.1 \pm 1.4$ | $76.6 \pm 1.0$ | $\mathbf{76.6 \pm 0.9}$ |
| | IncepGCN | $76.7 \pm 1.7$ | $77.4 \pm 1.5$ | $\mathbf{77.6 \pm 1.1}$ | $77.5 \pm 1.3$ | $77.3 \pm 1.2$ | $\mathbf{77.9 \pm 1.0}$ |
| Physics | GCN | $90.2 \pm 0.9$ | $91.6 \pm 0.8$ | $\mathbf{91.9 \pm 0.7}$ | $83.5 \pm 2.2$ | $\mathbf{86.3 \pm 1.8}$ | $85.2 \pm 2.3$ |
| | JKNet | $90.6 \pm 1.7$ | $91.0 \pm 0.9$ | $\mathbf{91.5 \pm 0.4}$ | $90.4 \pm 1.5$ | $91.5 \pm 0.7$ | $\mathbf{91.7 \pm 0.6}$ |
| | IncepGCN | $91.0 \pm 1.1$ | $91.4 \pm 0.5$ | $\mathbf{91.5 \pm 0.3}$ | $90.5 \pm 0.8$ | $91.4 \pm 0.8$ | $\mathbf{91.5 \pm 0.6}$ |

| Dataset | Backbone Finite | 16-layer Original | DropEdge | C-DropEdge | 32-layer Original | DropEdge | C-DropEdge |
|---|---|---|---|---|---|---|---|
| Cora | GCN | $36.3 \pm 13.8$ | $55.1 \pm 5.2$ | $\mathbf{58.5 \pm 3.9}$ | $20.1 \pm 2.4$ | $22.1 \pm 2.0$ | $\mathbf{24.7 \pm 1.8}$ |
| | JKNet | $79.9 \pm 1.6$ | $82.2 \pm 0.7$ | $\mathbf{82.4 \pm 0.8}$ | $80.4 \pm 1.4$ | $81.1 \pm 0.5$ | $\mathbf{82.6 \pm 0.9}$ |
| | IncepGCN | $78.7 \pm 1.0$ | $80.2 \pm 1.3$ | $\mathbf{82.0 \pm 0.8}$ | $78.5 \pm 1.8$ | $80.9 \pm 0.8$ | $\mathbf{81.6 \pm 0.9}$ |
| Citeseer | GCN | $30.8 \pm 2.2$ | $35.7 \pm 1.9$ | $\mathbf{36.6 \pm 2.0}$ | $21.7 \pm 3.0$ | $23.1 \pm 1.0$ | $\mathbf{25.2 \pm 0.9}$ |
| | JKNet | $69.2 \pm 1.2$ | $68.8 \pm 1.6$ | $\mathbf{69.5 \pm 1.0}$ | $68.1 \pm 2.3$ | $69.9 \pm 1.4$ | $\mathbf{70.1 \pm 0.6}$ |
| | IncepGCN | $68.4 \pm 1.2$ | $70.7 \pm 1.0$ | $\mathbf{71.0 \pm 1.0}$ | $68.6 \pm 1.9$ | $70.2 \pm 0.8$ | $\mathbf{70.7 \pm 0.9}$ |
| Pubmed | GCN | $39.5 \pm 10.3$ | $37.0 \pm 9.6$ | $\mathbf{42.6 \pm 6.4}$ | $36.3 \pm 8.4$ | $37.4 \pm 8.8$ | $\mathbf{38.5 \pm 6.1}$ |
| | JKNet | $76.6 \pm 0.9$ | $76.1 \pm 0.8$ | $\mathbf{76.9 \pm 0.9}$ | $77.1 \pm 0.8$ | $77.0 \pm 0.9$ | $\mathbf{77.2 \pm 1.0}$ |
| | IncepGCN | $76.5 \pm 1.5$ | $76.7 \pm 1.3$ | $\mathbf{77.2 \pm 1.0}$ | $77.0 \pm 1.4$ | $77.2 \pm 1.3$ | $\mathbf{77.8 \pm 1.0}$ |
| Physics | GCN | $41.6 \pm 6.2$ | $45.8 \pm 6.3$ | $\mathbf{46.2 \pm 5.7}$ | $28.8 \pm 9.4$ | $31.1 \pm 8.8$ | $\mathbf{36.2 \pm 8.4}$ |
| | JKNet | $90.3 \pm 1.0$ | $91.2 \pm 0.5$ | $\mathbf{91.5 \pm 0.4}$ | $90.6 \pm 1.0$ | $91.3 \pm 0.8$ | $\mathbf{91.6 \pm 0.6}$ |
| | IncepGCN | $91.4 \pm 0.4$ | $92.0 \pm 0.5$ | $\mathbf{92.0 \pm 0.4}$ | OOM | OOM | OOM |

OOM means Out-Of-Memory.

## G.2 PERFORMANCE OF USING DIFFERENT BACKBONES

The results presented in Table 1 are the best across different backbones. Table 5 further compares the results of vanilla GNN, DropEdge and C-DropEdge for three different backbones (GCN, JKNet and IncepGCN). From the table we can conclude that Critical DropEdge consistently outperforms DropEdge and vanilla GNNs.

## G.3 TRAINABILITY OF GNTK WITH CRITICAL DROPEDGE

We implement deep GCNs with/without DropEdge and C-DropEdge to compare the training performance. The results are shown in Figure 4. The training loss of GCN and GCN with DropEdge has a slower convergence rate and converges to a larger error compared to GCN with C-DropEdge. This indicates that C-DropEdge can achieve better trainability compared to GCN and GCN with DropEdge. Besides, the convergence results of GNTKs with C-DropEdge are shown in Figure 5. Compared to residual connection, we find that the normalized GNTK elements would not converge to a single value, and the convergence rate curve does not depend on the depth. This implies that residual connection mitigates the trainability loss by slowing down the exponential convergence rate, while C-DropEdge directly changes the convergence results.

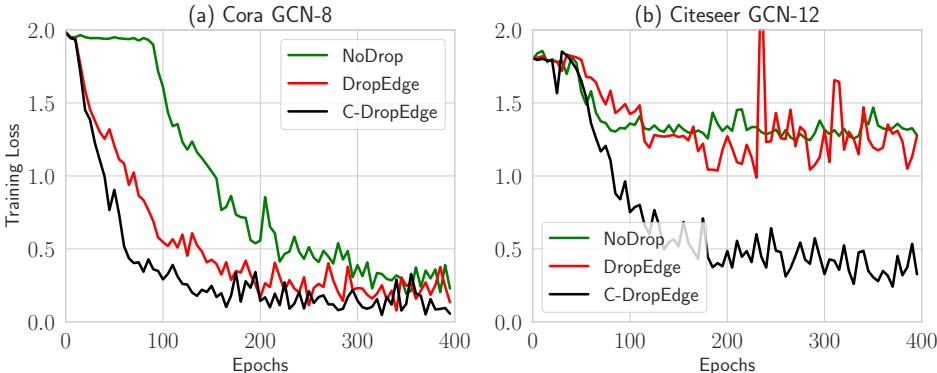

Figure 4: Training loss as a function of epochs. (a) We implement 8-layer GCN, GCN with DropEdge, GCN with C-DropEdge on Cora. (b) 12-layer GCN, GCN with DropEdge, GCN with C-DropEdge on Citeseer. The training loss of GCN and GCN with DropEdge has a slower convergence rate and converges to a larger error compared to GCN with C-DropEdge.

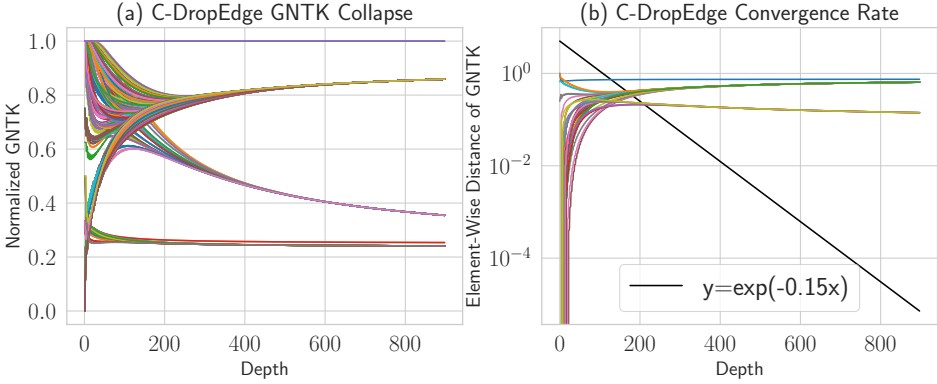

Figure 5: Convergence rate of GNTK with C-DropEdge. (a) Elements of the normalized (residual connection) GNTK as a function of the depth, defined as $Rl + r$. Different elements tend to have different value as depth grows (b) The element-wide distance of the normalized GNTK as a function of the depth. The converge rate is no longer bounded by an exponential function. Instead, it remains parallel to the horizontal depth axis.

