# OpenReview forum: "Towards Deepening Graph Neural Networks: A GNTK-based Optimization Perspective"
_ICLR.cc/2022/Conference — ICLR 2022 Poster_

### Official Review · Reviewer_KNdY · 2021-10-29

**Correctness:** 3
**Technical Novelty And Significance:** 3
**Empirical Novelty And Significance:** 2
**Recommendation:** 6
**Confidence:** 3

**Main Review:**

Strength:
(a) They are the first to focus on the trainability of deep GCNs through GNTK, instead of expressive power.
(b) Analysis and proof of the exponential convergence rate of the GNTK and that of residual techniques.

Weakness:
 (a) An error in and equation 6 when formulating GCNs. Graph Convolutional Networks do not have such decoupled formulations.
 (b) A fundamental error in and equation 1, 7 when formulating GNNs or GCNs. The non-linear activation function phi appears at the most INNER place in equation 1, 7, before the transformation. But neural networks usually have their activation after the transformation. Note that the activation functions are NON-LINEAR functions. All the analyses based on such an equation may be rechecked.
(c) Their experiments are based on only three citation datasets, which are limited both in fields and scales.
(d) No detailed information such as data splits is provided.


**Summary Of The Paper:**

They first investigate the trainability of deep GCNs through GNTK, instead of expressive power, and theoretically prove that the convergence rate of the GNTK is exponential. A similar analysis is given to residual techniques in GCNs, which shows residual techniques can alleviate such a problem to some extent but cannot solve it. Finally, based on their findings and analysis, they propose an edge-based sampling method named Critical DropEdge, to overcome the exponential decay of trainability.

**Summary Of The Review:**

The paper's insights are good, but the theoretical analysis seems based on a wrong equation, so they may need to recheck the results. Experiments conducted on more types of datasets in different fields and scales are required.

---

> ### Author Response · Authors · 2021-11-21
> **Reply to Reviewer KNdY**
>
> Thank you for constructive comments! We hope our answers below address all your concerns.
>
> ### 1. An error in and equation 6 when formulating GCNs. Graph Convolutional Networks do not have such decoupled formulations.
>
> Equation 6 can be equivalently written in the matrix formation, $H^{(l)}\_{(0)} = \tilde{D}^{-1}\tilde{A} H^{(l-1)}\_{(R)}$, where $\tilde{A} = A +I$ is the adjacency matrix with self-loop, and $\tilde{D}^{-1} \tilde{A}$ is the normalized adjacency matrix. Equation 6 in our paper is in the formation of node representation.
>
> ### 2. A fundamental error in and equation 1, 7 when formulating GNNs or GCNs. The non-linear activation function phi appears at the most INNER place in equation 1, 7, before the transformation. But neural networks usually have their activation after the transformation. Note that the activation functions are NON-LINEAR functions. All the analyses based on such an equation may be rechecked.
>
> For multilayer perceptron (Equations 1, 7), the different order of weight and activation function is basically the same, the only difference is the first layer and the last layer. In particular, there are two ways to write Equation 1,
>
> (1) $h^{(l)} = \phi(y^{(l)})$ and $y^{(l)}= W^{(l)}h^{(l-1)} + b^{(l)}$
>
> (2) $h^{(l)} = W^{(l)}y^{(l-1)} + b^{(l)}$ and $y^{(l)}= \phi(h^{(l)})$
>
> In (1) the $h$ is named as post-activation, and in (2) $h$ is called pre-activation. For pure MLP without aggeration, the resulting behavior of NTK in the large depth does not rely on the way we express MLP [1] (see Eq 1 in the paper).
>
> For the graph network, we investigate the GNTK in a large depth. The only difference with pure MLP is that we add an aggregation operation (Eq 6 in our manuscript). Because aggravation and MLP are alternate, the result does not depend on the order of activation and weights in MLP. To align with the standard way of writing GCN, we have modified Equations 1 and 7.
>
> ### 3. The experiments are based on only three citation datasets, which are limited both in fields and scales.
>
> We apply our Critical DropEdge to a larger and denser graph named Physics [1], which is a co-author network in Physics domain. The information is listed as follows,
>
> |Dataset | Nodes  |Edges | Classes | Features| Critical Edge sampling| Train/Val/Test |
> |---|---|--|---|---|---|------|
> |Physics|34493|247962|5|500| 6.96%|0.003/0.004/0.99|
>
>
> More details regarding this dataset can be found in Table 3 and Appendix E.1. The average node degree is 7.2. The critical edge sampling percentage is $6.96 \%$ accordingly. The comparison results are summarized as follows, from which we find that Critical DropEdge method can still outperform counterpart methods in a denser graph.
>
> |Backbone | Method   |4-layer| 8-layer | 16-layer| 32-layer|
> |---|-----|------|-------|-----|------|
> |GCN|Original| 90.2 ± 0.9|83.5 ± 2.2|41.6 ± 6.2|  28.8 ± 9.4|
> |GCN |DropEdge|  91.6 ± 0.8|**86.3 ± 1.8**|  45.8 ± 6.3| 31.1 ± 8.8|
> |GCN|C-DropEdge| **91.9 ± 0.7**| 85.2 ± 2.3| **46.2 ± 5.7**|**36.2 ± 8.4**|
> | |
> |JKNet|Original| 90.6 ± 1.7|90.4 ± 1.5|90.3 ± 1.0| 90.6 ± 1.0|
> |JKNet|DropEdge|  91.0 ± 0.9 |91.5 ± 0.7|  91.2 ± 0.5| 91.3 ± 0.8|
> |JKNet|C-DropEdge| **91.5 ± 0.4**|**91.7 ± 0.6**| **91.5 ± 0.4**|**91.6 ± 0.6**|
> ||
> |IncepGCN|Original| 91.0 ± 1.1|90.5 ± 0.8|91.4 ± 0.4| OOM|
> |IncepGCN|DropEdge| 91.4 ± 0.5 | 91.4 ± 0.8|   92.0 ± 0.5| OOM|
> |IncepGCN|C-DropEdge| **91.5 ± 0.3**|**91.5 ± 0.6**| **92.0 ± 0.4**|OOM|
>
> OOM means Out-Of-Memory
>
> [1] Xiao, Lechao, Jeffrey Pennington, and Samuel Schoenholz. "Disentangling trainability and generalization in deep neural networks." International Conference on Machine Learning. PMLR, 2020.
>
> [2] Shchur O, Mumme M, Bojchevski A, Günnemann S. Pitfalls of graph neural network evaluation. arXiv preprint arXiv:1811.05868. 2018 Nov 14.

---

> > ### Comment · Reviewer_KNdY · 2021-12-03
> > **Thank you for your detailed response**
> >
> > My concerns are well addressed. I would like to raise my scores.

---

> ### Author Response · Authors · 2021-11-29
> **We would love to hear your feedback on our rebuttal**
>
> Dear Reviewer KNdY,
>
> As the discussion period is close to the end and we have not yet heard back from you, we wanted to reach out to see if our rebuttal response has addressed your concerns.
>
> We are more than happy to discuss further if you have any further concerns and issues, please kindly let us know your feedback. Thank you for your time and help!

---

### Official Review · Reviewer_UKok · 2021-10-30

**Correctness:** 3
**Technical Novelty And Significance:** 3
**Empirical Novelty And Significance:** 2
**Recommendation:** 5
**Confidence:** 3

**Main Review:**

The theoretical perspective presented is interesting. However, I have the following comments:

1) In Corollary 1 it is claimed that there exists some L_0 for which for all L > L_0 the GNTK at depth L is singular. In the proof of the corollary, the authors say that this follows from Theorem 1. However, I do not see exactly why this is the case. The fact that the limit of the GNTK is singular does not imply that there exist some finite L_0 for which this is singular. Is this right?

2) In Theorem 2, I presume that there is some implicit relation between \tilde{\lambda_2} and \tilde{\alpha}, otherwise the last sentence about the relation of the second eigenvalue of \mathcal{A}(G) is irrelevant to the rest of the result and to the discussion after the Theorem. In other words, for the Theorem 2 to imply that “residual connection can slow down convergence rate”, I imagine that the authors are referring to the fact that \tilde{\alpha} in Theorem 2 is larger than \alpha in Theorem 1, but this is never stated.

3) The \alpha in Theorem 3 has to be the same as the \alpha in Theorem 1 for the story to make sense. I understand that this is the case, but this is never mentioned.

4) The solution proposed by the authors not to converge to a singular GNTK is to break the connectivity in the graph by dropping edges. However, this brings up a few natural follow-up concerns that are not addressed:
i) Wouldn’t you still have the same problem within each connected component of the new graph?
ii) Is really destroying information (deleting edges) present in your graph the best way to shield against the effect of deeper networks?
iii) Can something be said either theoretically or empirically about the GNTK of such networks? One would expect a Theorem like 1, 2, or 3 but for this new case showing that in the limit this architecture converges to something useful for classification, but there is no discussion in this direction.

5) The empirical results of Critical-DropEdge are not too much better than the baseline which suffers from exponential convergence. For example, the vanilla GCN in CORA goes from an accuracy of 79.8% with 4 layers to 20.1% with 32 layers, a massive decay. However, with the technique presented here this drop goes from 82.0% to 24.7%, which is in the same order of a massive drop. Moreover, for GCN, even with the proposed critical DropEdge, the architecture with 4 layers outperforms all of the deeper alternatives across the three datasets studied. In this sense, it seems like the proposed methodology mildly slows down the deterioration with depth, but cannot leverage depth since shallow GCNs are still performing better.

6) There are several typos and inconsistencies that should be addressed. E.g., if we only focus on page 6:
- in Theorem 2 the first appearance of \alpha should be \tilde{\alpha}
- in Theorems 2 and 3 the authors refer to v and v’ as matrices when they are vectors
- the probability p=|E|/E_t is not the probability of dropping an edge but the probability of retaining an edge.
- the sentence “the probability that there is an open path from some fixed point (say the origin) to a distance decreases polynomially” seems to be taken from somewhere else and copied out of context. What is an “open path” here? What is the “origin” in a graph? What “distance” are we referring to?
- Maybe it might be better to cite the original paper from Erdos-Renyi using the actual date. This is a paper from the 60s and it might be misleading to present this result as something from 2011 even if this paper was included in some more modern collection of papers.

**Summary Of The Paper:**

This paper uses a GNTK formulation to show how the performance of a deep GCN decays for node classification tasks with increasing depth. The authors show that in the infinite-width settings (where NTKs are valid) a very deep network amounts to a constant GNTK, meaning that the kernel value between any pair of nodes converges to the same value. Consequently, node classification performance degrades drastically with depth. Based on this analysis, the authors then propose a workaround to mitigate this decay by dropping edges in the original graph.


**Summary Of The Review:**

The authors present an interesting perspective to justify the decay in performance of deep GCNs. On the downside, the provided solution is not studied within the same theoretical framework and does not show promising empirical results.

---

> ### Author Response · Authors · 2021-11-21
> **Reply to Reviewer UKok Part III**
>
>
>
> ### References
>
> [1] Rosenthal JS. Convergence rates for Markov chains. Siam Review. 1995 Sep;37(3):387-405.
>
> [2] Erdős P, Rényi A. On the strength of connectedness of a random graph. Acta Mathematica Hungarica. 1961 Mar 1;12(1):261-7.
>
> [3] Ding, Xue, and Tiefeng Jiang. "Spectral distributions of adjacency and Laplacian matrices of random graphs. ``The annals of applied probability (2010): 2086-2117.
>
> [4] Chung, Fan, and Mary Radcliffe. ``On the spectra of general random graphs." the electronic journal of combinatorics (2011): P215-P215.
>
> [5] Zhou K, Huang X, Li Y, Zha D, Chen R, Hu X. Towards deeper graph neural networks with differentiable group normalization. arXiv preprint arXiv:2006.06972. 2020 Jun 12.
>
> [6] Shchur O, Mumme M, Bojchevski A, Günnemann S. Pitfalls of graph neural network evaluation. arXiv preprint arXiv:1811.05868. 2018 Nov 14.

---

> ### Author Response · Authors · 2021-11-21
> **Reply to Reviewer UKok Part II**
>
> ### 5. The empirical results of Critical-DropEdge are not too much better than the baseline which suffers from exponential convergence. It seems like the proposed methodology mildly slows down the deterioration with depth, but cannot leverage depth since shallow GCNs are still performing better.
>
> Our closest baseline is DropEdge. From Table 5 in our revised manuscript, we find that our C-DropEdge can outperform DropEdge and original backbones. Besides, with the help of residual connection (JKNet) or spectral techniques (InceptionGCN), our C-DropEdge can achieve the same performance in deep GNN (32-layer) as shallow network (4-layer). This can be found in Table 1 as follows,
>
> |Dataset|Methods|4-layer|8-layer|16-layer|32-layer|
> |---|----|----|----|---|---|
> | Cora| GCN| 79.8 ± 1.1| 73.2 ± 2.7 | 36.3 ± 13.8| 20.1 ± 2.4|
> |Cora|DropEdge| 82.2 ± 0.7| 82.0 ± 0.9| 82.2 ± 0.7| 82.1 ± 0.5|
> |Cora|DGN| 82.0 ± 0.9| 80.2 ± 0.8| 77.7 ± 1.0| 73.0 ± 0.8|
> |Cora|C-DropEdge| **82.5 ± 0.7**| **82.3 ± 0.6**| **82.4 ± 0.8**| **82.6 ± 0.9**|
> | |
> |Citeseer|GCN| 61.2 ± 3.0| 50.2 ± 5.7| 30.8 ± 2.2| 21.7 ± 3.0|
> |Citeseer|DropEdge| 70.2 ± 1.0| 70.8 ± 1.1| 70.7 ± 1.0| 70.2 ± 0.8|
> |Citeseer|DGN| 69.0 ± 0.9| 66.5 ± 1.1| 62.9 ± 1.2| 63.2 ± 0.9|
> |Citeseer|C-DropEdge|**70.8 ± 0.6**| **70.9 ± 0.9**| **71.0 ± 1.0**| **70.7 ± 0.9**|
> | |
> |Pubmed|GCN|77.4 ± 0.7| 57.2 ± 8.4| 39.5 ± 10.3| 36.3 ± 8.4|
> |Pubmed|Dropedge |77.6 ± 1.4| 77.3 ± 1.3| 76.7 ± 1.3| 77.2 ± 1.3|
> |Pubmed|DGN |**78.2 ± 1.0**| 77.8 ± 1.2| 77.2 ± 1.3| 77.0 ± 1.1|
> |Pubmed|C-DropEdge| 78.0 ± 0.4| **77.9 ± 1.0**| **77.2 ± 1.0**| **77.8 ± 1.0**|
> | |
> |Physics|GCN| 90.2 ± 0.9| 83.5 ± 2.2| 41.6 ± 6.2| 28.8 ± 9.4|
> |Physics|Dropedge| 91.6 ± 0.8| 91.5 ± 0.7| 91.2 ± 0.5 |91.3 ± 0.8|
> |Physics|DGN|**92.2 ± 1.0**| 86.4 ± 0.7| 83.4 ± 0.6| 83.2 ± 0.8|
> |Physics|C-DropEdge| 91.9 ± 0.7| **91.7 ± 0.6**| **92.0 ± 0.4**| **91.6 ± 0.6**|
>
> We have introduced an additional baseline method named DGN [5] which is a normalized-based approach. It utilizes differentiable group normalization which to overcome the over-smoothing problem. The backbone of DGN includes GCN and GAT.
>
> Besides, we  introduced a larger and denser graph named Physics [6], which is a co-author network in Physics domain. The information is listed as follows,
>
> |Dataset | Nodes  |Edges | Classes | Features| Critical Edge sampling| Train/Val/Test |
> |---|---|--|---|---|---|------|
> |Physics|34493|247962|5|500| 6.96%|0.003/0.004/0.99|
>
> All the experiments are conducted in the semi-supervised setting, and we report the best performance across different backbones for DropEdge, C-DropEdge, and DGN.
>
> From the Table, we find that C-DropEdge can outperform both DropEdge and DGN.

---

> ### Author Response · Authors · 2021-11-21
> **Reply to Reviewer UKok Part I**
>
> Thank you for constructive comments! We hope our answers below address all your concerns.
>
> ### 1. In Corollary 1 it is claimed that there exists some $L_0$ for which for all $L > L_0$ the GNTK at depth $L$ is singular. However, I do not see exactly why this is the case.
>
> Theorem 1 tells us that the GNTK convergences to a singular matrix at an exponential rate, this formally implies that the distance between GNTK at $L_0$ and limiting GNTK $\Theta^{(\infty)}\_{(R)}(\mathcal{V},\mathcal{V})$ is bounded by an exponential order,
> $$
> \| \Theta^{(L\_0)}\_{(R)}(\mathcal{V},\mathcal{V}) - \Theta^{(\infty)}\_{(R)}(\mathcal{V},\mathcal{V}) \|_2 = O(\alpha^{L\_0})
> $$
>
> By the $\varepsilon-\delta$ limiting definition, we can say that, there exist a $L_0 >0$ for any $\varepsilon >0$, if $L > L\_0 $, then $\| \Theta^{(L\_0)}\_{(R)}(\mathcal{V},\mathcal{V}) - \Theta^{(\infty)}\_{(R)}(\mathcal{V},\mathcal{V}) \|_2 \le \varepsilon$.
>
> To simiplfy the demontration, we have modified the Corollary 1.
>
> ### 2.  I imagine that the authors are referring to the fact that $\tilde{\alpha}$ in Theorem 2 is larger than $\alpha$ in Theorem 1, but this is never stated.
>
> We appreciate this comment. According to [1], the relationship between convergence rate $\alpha$ and second largest eigenvalue $\lambda_2$ of transition matrix $\mathcal{A}(G)$ can be expressed as,
>
> $$
> \big|\Theta^{(l)}_{(r)}(u,u') - \vec{\pi}(G)^T  \big( R l \vec{v}  + \vec{v}'\big)\big| \le C l^{J-1} \lambda_2^{l-J} \le C  \alpha^l
> $$
>
> where $\alpha > \lambda_2$, and $J > 1$ is the size of the largest Jordan block of $\mathcal{A}(G)$. From the above inequality, we can conclude that the second largest eigenvalue almost govern the convergence rate. However, to maintain rigor, we cannot directly obtain $\alpha < \tilde{\alpha}$. Instead, from the result that $\tilde{\lambda}_2 > \lambda_2$, we say that $\tilde{\alpha}$ is roughly larger than $\alpha$.
>
> We have added this discussion to the manuscript.
>
> ### 3. The $\alpha$ in Theorem 3 has to be the same as the $\alpha$ in Theorem 1 for the story to make sense. I understand that this is the case, but this is never mentioned.
>
> Thanks for pointing it out. we have added a discussion on this.
>
> ### 4. GNTK with Critical DropEdge brings up a few natural follow-up concerns that are not addressed: i) Wouldn’t you still have the same problem within each connected component of the new graph? ii) Is really destroying information (deleting edges) present in your graph the best way to shield against the effect of deeper networks? iii) Can something be said either theoretically or empirically about the GNTK of such networks? One would expect a Theorem like 1, 2, or 3 but for this new case showing that in the limit this architecture converges to something useful for classification, but there is no discussion in this direction.
>
> Thank you for raising these concerns, we answer your questions one by one.
>
> (i) According to the critical percolation theory, there exists a large connected component of order $O(n)$ [2], here $n$ is the number of nodes in a graph, when we use the critical drop rate. In the thermodynamic limit, or equivalently, $n \rightarrow \infty$, we have an infinite number of connected components, and there exists an infinitely large of $O(n^{3/2})$, which implies that the convergence of GNTK is not simply exponential. It would be interesting to consider achieving some theoretical results for the limiting behavior of the GNTK in the large depth with Critical DropEdge. Since the Critical DropEdge is originated from the random graph, a promising way is to analyze the spectral distributions of adjacency and Laplacian matrices with a random graph model, like existing studies [3, 4].
>
> (ii) We would like to emphasize that C-DropEdge will not drop information. Instead, to preserve information, edges will be resampled at every iteration. This implies that at each iteration during training, the GNN takes a random and large graph. For the whole training process, we can think that we have used all the information on the graph.
>
> (iii) The convergence results of GNTKs with C-DropEdge are shown in Figure 5. Compared to residual connection, we find that the normalized GNTK elements will not converge to a single value, and the convergence rate curve does not depend on the depth. This implies that residual connection mitigates the trainability loss by slowing down the exponential convergence rate, while C-DropEdge directly changes the convergence results.
>
> We have added a discussion regarding this for the readers of interest.

---

> > ### Comment · Reviewer_UKok · 2021-11-21
> > **A quick follow-up on the first two points**
> >
> > 1. I can be epsilon away from a singular matrix for arbitrary small epsilon > 0 and not be singular, right?
> >
> > E.g., consider the sequence of matrices {A_n} where A_n = B + (1/n) C, with B singular and C full rank.
> > Then, it is clear that A_\inf is singular BUT there is no n_0 such that for all finite n > n_0 we have that A_n is singular.
> >
> > I don't see here how an epsilon-delta argument really solves this issue.
> >
> > 2. What does "roughly larger" mean? It seems that you are depending on Theorem 2 to justify the empirical observation that residual connections slow down the convergence rate, but you actually have no precise claim relating the convergence bounds in Theorems 1 and 2 if you only mention that one is "roughly larger" than the other.

---

> > > ### Author Response · Authors · 2021-11-23
> > > **Thanks for your comments**
> > >
> > > Thank you for your comments.
> > >
> > > #### 1. Limiting behavior of GNTK
> > >
> > > Let us slightly modify the your example, which is closer to our theorem. Consider the sequence of matrices $\{A_n\}$ where $A_n =  B + \alpha^n C$ ($0<\alpha<1$), with B singular and C full rank. Then we have, $\det(B) = 0$ and $\det(C) \neq 0$, here $\det$ is the determinant of a square matrix. By looking at $\det(A_n)$, we can say that there exists a $n_0$ for any $\varepsilon > 0$, if $n > n_0$ then $\det(A_n) < \varepsilon $.
> > >
> > > We understand what you mean is that a matrix with a determinant close to 0 is not a singular matrix in the strict sense. Our original intention is to show that GNTK tends to the singular matrix very fast (at an exponential rate), and when $L_0$ is large, then $\alpha^{L_0}$ is extremely small, the corresponding GNTK has begun to show functional problems. For example, the determinant is close to 0, and there is an eigenvalue close to 0, which will cause trainability problems of corresponding GNNs. This can correspond to the experiment results of train and test accuracy depending on the depth shown in Figure 2 in our manuscript.
> > >
> > > We hope this explanation can clarify the problem.
> > >
> > > #### 2. Relationship between $\alpha$ in Theorem 1 and $\tilde{\alpha}$ in Theorem 2.
> > >
> > > We can claim that $\alpha \in [\lambda_2,1)$ and $\tilde{\alpha} \in [\tilde{\lambda}\_2,1)$, based on the inequliaty $\big|\Theta^{(l)}_{(r)}(u,u') - \vec{\pi}(G)^T  \big( R l \vec{v}  + \vec{v}'\big)\big| \le C l^{J-1} \lambda_2^{l-J} \le C  \alpha^l$ [1]
> > >
> > > Furthermore, according to Theorem 2, we have $\tilde{\lambda}_2 > \lambda_2$ which causes $\tilde{\alpha}$ has a smaller value range. Thus we can say that $\tilde{\alpha}$ tends to be greater than $\alpha$.
> > >
> > > [1] Rosenthal JS. Convergence rates for Markov chains. Siam Review. 1995 Sep;37(3):387-405.

---

> ### Author Response · Authors · 2021-11-29
> **We would love to hear more feedback on our rebuttal**
>
> Dear Reviewer UKok,
>
> Thanks for your quick follow-up on the first two points, we wanted to reach out to see if our rebuttal response has addressed all your concerns.
>
> We are more than happy to discuss further if you have any further concerns and issues, please kindly let us know your feedback. Thank you for your time and help!

---

### Official Review · Reviewer_sX4L · 2021-11-01

**Correctness:** 3
**Technical Novelty And Significance:** 2
**Empirical Novelty And Significance:** 2
**Recommendation:** 6
**Confidence:** 2

**Main Review:**

1. Previous methods only explore the expressivity of GCN and this work focuses on a more fundamental problem, whether gradient descent can find a good solution for deepGCN.

2. This paper has a solid and detailed mathematical derivation of the proposed theory. The experiments actually show that the residual connection doesn’t solve the problem but slows down the collapse speed during training and the distance between elements drops exponentially, which corresponds to the exponential decay of trainability.

Comments:
1. The proposed solution edge drop has been widely applied in many models. The optimal dropout rate is not a significant breakthrough. It could be easily found by the hyperparameters search.

2. The experimental setting still has some problems. There is only one baseline which is published in 2020. Maybe the author could incorporate more methods that aim to train a deep graph convolutional model to show their solution's superior performance.

3. The comparison is not fair. The hyperparameters for the baseline have been set the same as the original paper. But the train/validation/test set may not be the same. Because this paper reported results for the baseline are not consistent with the original paper. For example for 8 layers, the Cora dataset reported by the original paper on GCN and JKNet model is 85.80 and 87.80. The corresponding result in this paper is 75.1,  82.0.


**Summary Of The Paper:**

This paper utilizes the latest technology called GNTK to explore the trainability of deep graph convolutional networks. They prove that the trainability will decay exponentially as layers go deeper and residual connection only mitigates this phenomenon. They proposed to utilize drop edge methods to solve this problem. The dropout rate is determined by the theory. And the experimental result shows that their solution is better than the baseline.

**Summary Of The Review:**

The experimental results prove the consistency between the theory and reality.
For the weakness of this paper, the proposed solutions to solve this problem are far less attractive and the comparison with the baseline is not fair.

---

> ### Author Response · Authors · 2021-11-21
> **Reply to Reviewer sX4L**
>
> Thank you for constructive comments! We hope our answers below address all your concerns.
>
> ### 1. The proposed solution edge drop has been widely applied in many models. The optimal dropout rate is not a significant breakthrough. It could be easily found by the hyperparameters search.
>
> The comparison between C-DropEdge and DropEdge shows that our critical DropEdge methods not only outperforms DropEdge, but also achieves more robust results. The whole search space for edge preserve percent is continuous, and if one wants to use grid search, for instance, diving the search space into 20 parts, with 5\% as a unit interval, then it will take 20 times longer than C-DropEdge.
>
> ### 2. There is only one baseline which is published in 2020. Maybe the author could incorporate more methods that aim to train a deep graph convolutional model to show their solution's superior performance.
>
> Thanks for your suggestion. We have introduced an additional baseline method named DGN [1] which is a normalized-based approach. It utilizes differentiable group normalization which to overcome the over-smoothing problem. The comparison results of our method with baseline methods including DGN, GCN, and DropEdge are listed as follows and in Table 1 in our revised manuscript. The backbone of DGN includes GCN and GAT.
>
> |Dataset|Methods|4-layer|8-layer|16-layer|32-layer|
> |---|----|----|----|---|---|
> | Cora| GCN| 79.8 ± 1.1| 73.2 ± 2.7 | 36.3 ± 13.8| 20.1 ± 2.4|
> |Cora|DropEdge| 82.2 ± 0.7| 82.0 ± 0.9| 82.2 ± 0.7| 82.1 ± 0.5|
> |Cora|DGN| 82.0 ± 0.9| 80.2 ± 0.8| 77.7 ± 1.0| 73.0 ± 0.8|
> |Cora|C-DropEdge| **82.5 ± 0.7**| **82.3 ± 0.6**| **82.4 ± 0.8**| **82.6 ± 0.9**|
> | |
> |Citeseer|GCN| 61.2 ± 3.0| 50.2 ± 5.7| 30.8 ± 2.2| 21.7 ± 3.0|
> |Citeseer|DropEdge| 70.2 ± 1.0| 70.8 ± 1.1| 70.7 ± 1.0| 70.2 ± 0.8|
> |Citeseer|DGN| 69.0 ± 0.9| 66.5 ± 1.1| 62.9 ± 1.2| 63.2 ± 0.9|
> |Citeseer|C-DropEdge|**70.8 ± 0.6**| **70.9 ± 0.9**| **71.0 ± 1.0**| **70.7 ± 0.9**|
> | |
> |Pubmed|GCN|77.4 ± 0.7| 57.2 ± 8.4| 39.5 ± 10.3| 36.3 ± 8.4|
> |Pubmed|Dropedge |77.6 ± 1.4| 77.3 ± 1.3| 76.7 ± 1.3| 77.2 ± 1.3|
> |Pubmed|DGN |**78.2 ± 1.0**| 77.8 ± 1.2| 77.2 ± 1.3| 77.0 ± 1.1|
> |Pubmed|C-DropEdge| 78.0 ± 0.4| **77.9 ± 1.0**| **77.2 ± 1.0**| **77.8 ± 1.0**|
> | |
> |Physics|GCN| 90.2 ± 0.9| 83.5 ± 2.2| 41.6 ± 6.2| 28.8 ± 9.4|
> |Physics|Dropedge| 91.6 ± 0.8| 91.5 ± 0.7| 91.2 ± 0.5 |91.3 ± 0.8|
> |Physics|DGN|**92.2 ± 1.0**| 86.4 ± 0.7| 83.4 ± 0.6| 83.2 ± 0.8|
> |Physics|C-DropEdge| 91.9 ± 0.7| **91.7 ± 0.6**| **92.0 ± 0.4**| **91.6 ± 0.6**|
>
> Here, we introduced a larger and denser graph named Physics [2], which is a co-author network in Physics domain. The information is listed as follows,
>
> |Dataset | Nodes  |Edges | Classes | Features| Critical Edge sampling| Train/Val/Test |
> |---|---|--|---|---|---|------|
> |Physics|34493|247962|5|500| 6.96%|0.003/0.004/0.99|
>
> All the experiments are conducted in the semi-supervised setting, and we report the best performance across different backbones for DropEdge, C-DropEdge, and DGN. From the Table, we find that C-DropEdge can outperform both DropEdge and DGN.
>
> ### 3. The comparison is not fair. The hyperparameters for the baseline have been set the same as the original paper. But the train/validation/test set may not be the same.
>
> For all methods including baselines and our methods, we use the semi-supervised setting for all datasets, and details are listed as follows and in (updated) Table 3.
>
> | Dataset| Nodes|Train/ Val /Test |
> |---|---|--|
> |Cora| 2,708|0.05/0.18/0.37|
> |Citeseer|3,327|0.04/0.15/0.30|
> |Pubmed|19,717|0.003/0.03/0.05|
> |Physics|34,493|0.003/0.004/0.99|
>
> For all dataset, we give 20 labels for each class in the training set.
>
> The Cora dataset reported by the original paper on GCN and JKNet model (85.80 and 87.80 for 8-layer) is in the full-supervised setting, and the hyperparameters for the DropEdge on semi-supervised learning can be found at GitHub: https://github.com/DropEdge/DropEdge.
>
> [1] Zhou K, Huang X, Li Y, Zha D, Chen R, Hu X. Towards deeper graph neural networks with differentiable group normalization. arXiv preprint arXiv:2006.06972. 2020 Jun 12.
>
> [2]  Shchur O, Mumme M, Bojchevski A, Günnemann S. Pitfalls of graph neural network evaluation. arXiv preprint arXiv:1811.05868. 2018 Nov 14.

---

> ### Author Response · Authors · 2021-11-29
> **We would love to hear your feedback on our rebuttal**
>
> Dear Reviewer sX4L,
>
> As the discussion period is close to the end and we have not yet heard back from you, we wanted to reach out to see if our rebuttal response has addressed your concerns.
>
> We are more than happy to discuss further if you have any further concerns and issues, please kindly let us know your feedback. Thank you for your time and help!

---

### Official Review · Reviewer_CDt6 · 2021-11-02

**Correctness:** 3
**Technical Novelty And Significance:** 3
**Empirical Novelty And Significance:** 3
**Recommendation:** 6
**Confidence:** 4

**Main Review:**

### Strengths

1. The investigation of the trainability of the ultra-wide GCNs via the GNTK view has its novelty for the graph machine learning community and provides new perspective for the over-smoothing problem.
2. The Proposition 1 provides more information on the power of the DropEdge method and motivates the derivation of C-DropEdge.
3. Detailed experimental analyses (Fig 1, Table 2) are provided.
4. Both the Sec 2 and the Sec 6 are well written and provide clear and fruitful preliminary knowledge on this work.

### Weaknesses

1. About the derivation of the C-DropEdge

   (1). The authors make extrapolation from the connected component size to information flow and show the information transforms in the critical random graph at a polynomial rate rather than an exponential rate. It would be better to further analyze the convergence rate of the GNTK in the critical random graph and show that the exponential drop of the trainability is mitigated by the proposed technique.

2. About the experimental results:

   (1). To validate that the C-DropEdge can better mitigate the loss of the trainability of deep GNNs, the training performance should be provided besides Table 1. Moreover, the convergence results of GNTKs with C-DropEdge should also be provided to compare the mitigating effect between residual connection and C-DropEdge.

   (2). Suggestions for the next version of this paper: there can be more baseline methods to verify the C-DropEdge. Different model architectures [1, 2] and normalization mechanisms [3, 4] can be tested to provide more general insight on the effect of the C-DropEdge for the methods in use in the graph ml community.

3. Minor issues:

   (1) In the theorem 2, the readers can only get that the original base $\alpha$ of the exponential convergence rate term is replaced by a new base $\tilde\alpha$ (it is even not defined), and the second largest eigenvalue of the transition matrix under the residual connection setting is larger. It is not straightforward to conclude that the "slow down" effect of the residual connection, especially for those who do not have enough theoretical background knowledge.

   (2) The wording "residual connection-resemble technique" is perplexing.

[1]. Chen et al, Simple and Deep Graph Convolutional Networks. ICML 2020

[2]. Li et al, Deepergcn: All you need to train deeper gcns. arxiv preprint.

[3]. Zhao et al, Pairnorm: Tackling oversmoothing in gnns. ICLR 2020.

[4]. Zhou et al, Towards Deeper Graph Neural Networks with Differentiable Group Normalization. NeurIPS 2020.

**Summary Of The Paper:**

This paper aims at tackling the well-known over-smoothing problem of deep graph neural networks (GNNs) from a theoretical perspective. Specifically, the authors exploit the Graph Neural Tangent Kernel (GNTK) and use it to analyze the trainablility of GNTK in the large depth, which provides insight for the reason of the over-smoothing problem. According to the theoretical analysis, the trainablility of wide and deep GNNs drops exponentially in the optimization process, and the residual connection can only mildly mitigate this problem. Based on the analysis, the authors modify the DropEdge method and propose the Critical DropEdge (C-DropEdge). Both numerical and real experiments are provided to demonstrate the validity of the theoretical analysis and the proposed C-DropEdge.

**Summary Of The Review:**

The over-smoothing problem of deep GNNs is a well-known problem in the graph ml community, which impedes the researchers to fully unlock the power of the GNNs due to the limits of the model capacity. The authors try to provide a new perspective via the GNTK view. However, there exists logical gap between the theoretical results and the claims and the derivation of the proposed C-DropEdge algorithms. Detailed experimental results are also necessary to better demonstrate the effect of the proposed method. **Thus, my recommendation for this paper is "weak rejection".** If the authors can address the questions in the above "Weakness" section well, I am willing to consider raising my scores.

********* After Rebuttal *********

The authors have addressed by concerns and I am willing to raise my scores to "Weak Accept".

---

> ### Author Response · Authors · 2021-11-21
> **Reply to Reviewer CDt6 Part II**
>
>
> ### 3. Suggestions for the next version of this paper: there can be more baseline methods to verify the C-DropEdge. Different model architectures and normalization mechanisms [3] can be tested.
>
> Thanks for your suggestion. We have introduced an additional baseline method named DGN [3] which is a normalized-based approach. It utilizes differentiable group normalization which to overcome the over-smoothing problem. The comparison results of our method with baseline methods including DGN, GCN and DropEdge are listed as follows and in Table 1 in our revised manuscript. The backbone of DGN includes GCN and GAT. All the experiments are conducted in the semi-supervised setting, and we report the best performance across different backbones for DropEdge, C-DropEdge, and DGN.
>
> |Dataset|Methods|4-layer|8-layer|16-layer|32-layer|
> |---|----|----|----|---|---|
> | Cora| GCN| 79.8 ± 1.1| 73.2 ± 2.7 | 36.3 ± 13.8| 20.1 ± 2.4|
> |Cora|DropEdge| 82.2 ± 0.7| 82.0 ± 0.9| 82.2 ± 0.7| 82.1 ± 0.5|
> |Cora|DGN| 82.0 ± 0.9| 80.2 ± 0.8| 77.7 ± 1.0| 73.0 ± 0.8|
> |Cora|C-DropEdge| **82.5 ± 0.7**| **82.3 ± 0.6**| **82.4 ± 0.8**| **82.6 ± 0.9**|
> | |
> |Citeseer|GCN| 61.2 ± 3.0| 50.2 ± 5.7| 30.8 ± 2.2| 21.7 ± 3.0|
> |Citeseer|DropEdge| 70.2 ± 1.0| 70.8 ± 1.1| 70.7 ± 1.0| 70.2 ± 0.8|
> |Citeseer|DGN| 69.0 ± 0.9| 66.5 ± 1.1| 62.9 ± 1.2| 63.2 ± 0.9|
> |Citeseer|C-DropEdge|**70.8 ± 0.6**| **70.9 ± 0.9**| **71.0 ± 1.0**| **70.7 ± 0.9**|
> | |
> |Pubmed|GCN|77.4 ± 0.7| 57.2 ± 8.4| 39.5 ± 10.3| 36.3 ± 8.4|
> |Pubmed|Dropedge |77.6 ± 1.4| 77.3 ± 1.3| 76.7 ± 1.3| 77.2 ± 1.3|
> |Pubmed|DGN |**78.2 ± 1.0**| 77.8 ± 1.2| 77.2 ± 1.3| 77.0 ± 1.1|
> |Pubmed|C-DropEdge| 78.0 ± 0.4| **77.9 ± 1.0**| **77.2 ± 1.0**| **77.8 ± 1.0**|
> | |
> |Physics|GCN| 90.2 ± 0.9| 83.5 ± 2.2| 41.6 ± 6.2| 28.8 ± 9.4|
> |Physics|Dropedge| 91.6 ± 0.8| 91.5 ± 0.7| 91.2 ± 0.5 |91.3 ± 0.8|
> |Physics|DGN|**92.2 ± 1.0**| 86.4 ± 0.7| 83.4 ± 0.6| 83.2 ± 0.8|
> |Physics|C-DropEdge| 91.9 ± 0.7| **91.7 ± 0.6**| **92.0 ± 0.4**| **91.6 ± 0.6**|
>
> From the Table, we find that C-DropEdge can outperform both DropEdge and DGN.
>
> ### 4. In the theorem 2, the readers can only get that the original base  of the exponential convergence rate term is replaced by a new base (it is even not defined), and the second largest eigenvalue of the transition matrix under the residual connection setting is larger.
>
> Thanks for pointing it out. According to [4], the relationship between convergence rate $\alpha$ and second largest eigenvalue $\lambda_2$ of transition matrix $\mathcal{A}(G)$ can be expressed as,
>
> $$
> \big|\Theta^{(l)}_{(r)}(u,u') - \vec{\pi}(G)^T  \big( R l \vec{v}  + \vec{v}'\big)\big| \le C l^{J-1} \lambda_2^{l-J} \le C  \alpha^l
> $$
>
> where $\alpha > \lambda_2$, and $J > 1$ is the size of the largest Jordan block of $\mathcal{A}(G)$. From the above inequality, we can conclude that the second largest eigenvalue almost govern the convergence rate and we can claim that $\alpha \in [\lambda_2,1)$ and $\tilde{\alpha} \in [\tilde{\lambda}\_2,1)$.
>
> Furthermore, according to Theorem 2, we have $\tilde{\lambda}_2 > \lambda_2$ which causes $\tilde{\alpha}$ has a smaller value range. Thus we can say that $\tilde{\alpha}$ tends to be greater than $\alpha$.
>
> We added this discussion to the revised submission.
>
> ### 5. The wording "residual connection-resemble technique" is perplexing.
>
> We have modified it to residual connection-based techniques. Please let us know if it is better.
>
> [1] Ding, Xue, and Tiefeng Jiang. "Spectral distributions of adjacency and Laplacian matrices of random graphs. ``The annals of applied probability (2010): 2086-2117.
>
> [2] Chung, Fan, and Mary Radcliffe. ``On the spectra of general random graphs." the electronic journal of combinatorics (2011): P215-P215.
>
> [3] Zhou K, Huang X, Li Y, Zha D, Chen R, Hu X. Towards deeper graph neural networks with differentiable group normalization. arXiv preprint arXiv:2006.06972. 2020 Jun 12.
>
> [4] Rosenthal JS. Convergence rates for Markov chains. Siam Review. 1995 Sep;37(3):387-405.

---

> ### Author Response · Authors · 2021-11-23
> **Reply to Reviewer CDt6 Part I**
>
> Thank you for constructive comments! We hope our answers below address all your concerns.
>
> ### 1. It would be better to further analyze the convergence rate of the GNTK in the critical random graph and show that the exponential drop of the trainability is mitigated by the proposed technique.
>
> Thanks for your suggestion. For the trainability of GCNs with Critical DropEdge, it would be interesting to consider achieving some results for the limiting behavior of the GNTK in the large depth. A promising direction is to analyze the spectral distributions of adjacency and Laplacian matrices, like existing studies [1, 2]. We believe it would take a lot of extra work to redesign the proof and develop novel techniques. Therefore, we leave the asymptotic analysis of GNTK for random graphs to future work.
>
> ### 2. The training performance should be provided besides Table 1. Moreover, the convergence results of GNTKs with C-DropEdge should also be provided.
>
> Thanks for your constructive suggestions. We have implemented deep GCN without/without DropEdge and C-DropEdge to compare the training performance. The result is shown in Figure 4 in our revised submission. Besides, the convergence results of GNTKs with C-DropEdge are shown in Figure 5. Compared to residual connection, we find that the normalized GNTK elements will not converge to a single value, and the convergence rate curve does not depend on the depth. This implies that residual connection mitigates the trainability loss by slowing down the exponential convergence rate, while C-DropEdge directly changes the convergence results.

---

> ### Author Response · Authors · 2021-11-29
> **We would love to hear your feedback on our rebuttal**
>
> Dear Reviewer CDt6,
>
> As the discussion period is close to the end and we have not yet heard back from you, we wanted to reach out to see if our rebuttal response has addressed your concerns.
>
> We are more than happy to discuss further if you have any further concerns and issues, please kindly let us know your feedback. Thank you for your time and help!

---

### Official Review · Reviewer_Bi6F · 2021-11-09

**Correctness:** 4
**Technical Novelty And Significance:** 3
**Empirical Novelty And Significance:** 3
**Recommendation:** 8
**Confidence:** 2

**Main Review:**

### Strengths:
1. The idea of using GNTK to understand the over-smoothing issue is interesting. I think this approach opens up new viewpoints and directions on understanding GNNs and should be interesting to the community.
2. The experiment results aligned with the theoretical claims and insights very well and showed that the messages in the paper are sound.
3. Dropping edges following the critical connectivity theorem is insightful. Although it is a sample add-on to the DropEdge method (Rong et al., 2019), this extra piece of information is important.

### Questions:
1. I am curious whether this critical drop rate would still be the best on some denser graphs? It seems that the three node classification benchmarks considered are all relative sparse graphs whose average node degree is less than 5. The critical drop rate leads to a graph whose average node degree is equal to 1. And this might be too sparse for some denser input graphs (i.e., dropping too much information)? I found this critical DropEdge also provides some insights on improving the Message Passing architecture. Thus some further exploration might be interesting.
2. The insights behind critical DropEdge is to break the connectivity. However, it is unclear how this would affect the theoretical results (Corollary 1). Do you think there are other possible methods to get rid of the exponential decay without affecting the connectivity?

### Weaknesses:
1. The baseline methods considered in the paper are limited. I agree that DropEdge should be the most relevant baseline, but it would be better to discuss some other methods (e.g., normalization-based approaches). I also find the related work section insufficient. It would be interesting if the new theoretical framework could provide some insights on the other research on understanding/solving the over-smoothing issue.

**Summary Of The Paper:**

This paper provided a novel viewpoint to understand the over-smoothing of deep GCNs by analyzing the asymptotic behavior of the Graph Neural Tangent Kernel (GNTK), which governs the optimization trajectory under gradient descent for wide GCNs. It is shown the trainability of GNTK (and thus wide and deep GCNs) drops at an exponential rate in the optimization process. It is also found that residual connection-resemble techniques can only help slow down the exponential decay of trainability but cannot overcome the problem. Following the "Critical Connectivity in Random Graph" by (Erdos & Renyi, 2011), the paper also provided insights on what critical edge-drop rate should be used in DropEdge. Experiments verify the theoretical results, and the critical DropEdge can consistently outperform other drop rates.

**Summary Of The Review:**

This paper tries to understand the over-smoothing problem of deep and wide GCNs using the theoretical framework provided by GNTKs. The results on the exponential decay of trainability, residual connections, and critical connectivity are insightful and should be interesting to the community. The experimental results are well-aligned with the claims and show the soundness of theoretical findings. Given the novelty and significance of the findings, I am happy to recommend this paper for acceptance.

---

> ### Author Response · Authors · 2021-11-21
> **Reply to Reviewer Bi6F Part II**
>
>
> ### 4. The baseline methods considered in the paper are limited.
>
> We have introduced an additional baseline method name DGN [5] which is a normalized-based approach. It utilizes differentiable group normalization which to overcome over-smoothing problem. The comparison results of our method with baseline methods including DGN, GCN and DropEdge are listed as follows and in Table 1 in our revised manuscript. The backbone of DGN includes GCN and GAT. All the experiments are conducted in the semi-supervised setting, and we report the best performance across different backbones for DropEdge, C-DropEdge, and DGN.
>
> |Dataset|Methods|4-layer|8-layer|16-layer|32-layer|
> |---|----|----|----|---|---|
> | Cora| GCN| 79.8 ± 1.1| 73.2 ± 2.7 | 36.3 ± 13.8| 20.1 ± 2.4|
> |Cora|DropEdge| 82.2 ± 0.7| 82.0 ± 0.9| 82.2 ± 0.7| 82.1 ± 0.5|
> |Cora|DGN| 82.0 ± 0.9| 80.2 ± 0.8| 77.7 ± 1.0| 73.0 ± 0.8|
> |Cora|C-DropEdge| **82.5 ± 0.7**| **82.3 ± 0.6**| **82.4 ± 0.8**| **82.6 ± 0.9**|
> | |
> |Citeseer|GCN| 61.2 ± 3.0| 50.2 ± 5.7| 30.8 ± 2.2| 21.7 ± 3.0|
> |Citeseer|DropEdge| 70.2 ± 1.0| 70.8 ± 1.1| 70.7 ± 1.0| 70.2 ± 0.8|
> |Citeseer|DGN| 69.0 ± 0.9| 66.5 ± 1.1| 62.9 ± 1.2| 63.2 ± 0.9|
> |Citeseer|C-DropEdge|**70.8 ± 0.6**| **70.9 ± 0.9**| **71.0 ± 1.0**| **70.7 ± 0.9**|
> | |
> |Pubmed|GCN|77.4 ± 0.7| 57.2 ± 8.4| 39.5 ± 10.3| 36.3 ± 8.4|
> |Pubmed|Dropedge |77.6 ± 1.4| 77.3 ± 1.3| 76.7 ± 1.3| 77.2 ± 1.3|
> |Pubmed|DGN |**78.2 ± 1.0**| 77.8 ± 1.2| 77.2 ± 1.3| 77.0 ± 1.1|
> |Pubmed|C-DropEdge| 78.0 ± 0.4| **77.9 ± 1.0**| **77.2 ± 1.0**| **77.8 ± 1.0**|
> | |
> |Physics|GCN| 90.2 ± 0.9| 83.5 ± 2.2| 41.6 ± 6.2| 28.8 ± 9.4|
> |Physics|Dropedge| 91.6 ± 0.8| 91.5 ± 0.7| 91.2 ± 0.5 |91.3 ± 0.8|
> |Physics|DGN|**92.2 ± 1.0**| 86.4 ± 0.7| 83.4 ± 0.6| 83.2 ± 0.8|
> |Physics|C-DropEdge| 91.9 ± 0.7| **91.7 ± 0.6**| **92.0 ± 0.4**| **91.6 ± 0.6**|
>
> From the Table, we find that C-DropEdge can outperform both DropEdge and DGN.
>
> ### 5.  I find the related work section insufficient.
>
> Thanks for your suggestion. We have updated some references in the section of Related Work. These references can be roughly divided into two classes, one is the regulation-based methods [6,7,8,9], and the other is a theoretical investigation on deep GNNs [10, 11, 12].
>
> [1] Shchur O, Mumme M, Bojchevski A, Günnemann S. Pitfalls of graph neural network evaluation. arXiv preprint arXiv:1811.05868. 2018 Nov 14.
>
> [2] Erdős P, Rényi A. On the strength of connectedness of a random graph. Acta Mathematica Hungarica. 1961 Mar 1;12(1):261-7.
>
> [3] Ding, Xue, and Tiefeng Jiang. "Spectral distributions of adjacency and Laplacian matrices of random graphs. ``The annals of applied probability (2010): 2086-2117.
>
> [4] Chung, Fan, and Mary Radcliffe. ``On the spectra of general random graphs." the electronic journal of combinatorics (2011): P215-P215.
>
> [5] Zhou K, Huang X, Li Y, Zha D, Chen R, Hu X. Towards deeper graph neural networks with differentiable group normalization. arXiv preprint arXiv:2006.06972. 2020 Jun 12.
>
> [6] Dwivedi, V. P., Joshi, C. K., Laurent, T., Bengio, Y., and Bresson, X. (2020). Benchmarking graph neural networks. arXiv preprint arXiv:2003.00982.
>
> [7] Zhao, Lingxiao, and Leman Akoglu. "Pairnorm: Tackling oversmoothing in gnns." arXiv preprint arXiv:1909.12223 (2019).
>
> [8] Zhou, Kaixiong, et al. "Towards deeper graph neural networks with differentiable group normalization." arXiv preprint arXiv:2006.06972 (2020).
>
> [9] Hou, Yifan, et al. "Measuring and improving the use of graph information in graph neural networks." International Conference on Learning Representations. 2019.
>
> [10] Cong, Weilin, Morteza Ramezani, and Mehrdad Mahdavi. "On Provable Benefits of Depth in Training Graph Convolutional Networks." arXiv preprint arXiv:2110.15174 (2021).
>
> [11] Loukas, Andreas. "What graph neural networks cannot learn: depth vs width." arXiv preprint arXiv:1907.03199 (2019).
>
> [12] Huang, Wenbing, et al. "Tackling Over-Smoothing for General Graph Convolutional Networks." arXiv preprint arXiv:2008.09864 (2020).

---

> > ### Comment · Reviewer_Bi6F · 2021-12-01
> > **Thanks for the detailed response**
> >
> > Thanks for your detailed response and sorry for my late reply. I have carefully read the response and the other reviewer's comments. I think all of my concerns have been addressed. Thus I would like to maintain my score.

---

> ### Author Response · Authors · 2021-11-23
> **Reply to Reviewer Bi6F Part I**
>
> Thank you for constructive comments! We hope our answers below address all your concerns.
> &nbsp;
>
> ### 1. Whether this critical drop rate would still be the best on some denser graphs?
>
> We apply our Critical DropEdge to a larger and denser graph named Physics [1], which is a co-author network in Physics domain. The information is listed as follows,
>
> |Dataset | Nodes  |Edges | Classes | Features| Critical Edge sampling| Train/Val/Test |
> |---|---|--|---|---|---|------|
> |Physics|34493|247962|5|500| 6.96%|0.003/0.004/0.99|
>
>
> More details regarding this dataset can be found in Table 3 and Appendix E.1. The average node degree is 7.2. The critical edge sampling percentage is $6.96 \%$ accordingly. The comparison results are summarized as follows, from which we find that Critical DropEdge method can still outperform counterpart methods in a denser graph.
>
> |Backbone | Method   |4-layer| 8-layer | 16-layer| 32-layer|
> |---|-----|------|-------|-----|------|
> |GCN|Original| 90.2 ± 0.9|83.5 ± 2.2|41.6 ± 6.2|  28.8 ± 9.4|
> |GCN |DropEdge|  91.6 ± 0.8|**86.3 ± 1.8**|  45.8 ± 6.3| 31.1 ± 8.8|
> |GCN|C-DropEdge| **91.9 ± 0.7**| 85.2 ± 2.3| **46.2 ± 5.7**|**36.2 ± 8.4**|
> | |
> |JKNet|Original| 90.6 ± 1.7|90.4 ± 1.5|90.3 ± 1.0| 90.6 ± 1.0|
> |JKNet|DropEdge|  91.0 ± 0.9 |91.5 ± 0.7|  91.2 ± 0.5| 91.3 ± 0.8|
> |JKNet|C-DropEdge| **91.5 ± 0.4**|**91.7 ± 0.6**| **91.5 ± 0.4**|**91.6 ± 0.6**|
> ||
> |IncepGCN|Original| 91.0 ± 1.1|90.5 ± 0.8|91.4 ± 0.4| OOM|
> |IncepGCN|DropEdge| 91.4 ± 0.5 | 91.4 ± 0.8|   92.0 ± 0.5| OOM|
> |IncepGCN|C-DropEdge| **91.5 ± 0.3**|**91.5 ± 0.6**| **92.0 ± 0.4**|OOM|
>
> OOM means Out-Of-Memory
>
> ### 2. Critical drop rate might be too sparse for some denser input graphs (i.e., dropping too much information)? I found this critical DropEdge also provides some insights on improving the Message Passing architecture.
>
> According to the critical percolation theory, there exists a large connected component of order $O(n)$ [2], here $n$ is the number of nodes in a graph, when we use the critical drop rate. To preserve information, edges will be resampled at every iteration. This implies that at each iteration during training, the GNN takes a random and large graph. For the whole training process, we can think that we have used all the information on the graph. Thanks for your suggestion on an interesting future exploration, we will consider working on improving the Message Passing architecture through critical DropEdge.
>
> ### 3. The insights behind critical DropEdge is to break the connectivity. However, it is unclear how this would affect the theoretical results (Corollary 1). Do you think there are other possible methods to get rid of the exponential decay without affecting the connectivity?
>
> It would be interesting to consider achieving some theoretical results for the limiting behavior of the GNTK in the large depth with Critical DropEdge. Since the Critical DropEdge is originated from random graph, a promising way is to analyze the spectral distributions of adjacency and Laplacian matrices with random graph model, like existing studies [3, 4]. We believe that there are other methods that overcome the exponential decay problem and Critical DropEdge is one of the way to solve it. The key idea is to modify the training dynamics of GNNs.

---

### Author Response · Authors · 2021-11-23
**Paper updated and further discussion**

Dear reviewers,

We have uploaded a paper revision addressing your concerns and suggestions, and making other improvements:

1. Simplified the condition for GNTK to be a singular matrix in Corollary 1.
2. Added a discussion on the relationship between convergence rate $\alpha$ and second-largest eigenvalue $\lambda_2$ of the transition matrix below Theorem 2.
3. Added a discussion on the convergence rate $\alpha$ below Theorem 3.
4. Added a further discussion of the relationship between Proposition 1 and the trainability of the corresponding GNTK in Appendix D.
5. Introduced a baseline method named DGN [1] which is a normalized-based approach and reported the results in Table 1.
6. Introduced a denser graph named Physics [2], and reported the results in Table 5.
7. Move original Table 1 to Table 5 in appendix F.2.
8. Added several related works on deep GNNs.
9. Added a discussion on the trainability of GNTK with Critical DropEdge in Appendix F.4 with experiments (Figure 4, Figure 5).
10. Corrected several typos.

All changes are marked with blue color.

We appreciate all reviewers for the hard work and helpful comments. We would like to address all reviewers’ concerns in the corresponding responses.

**References**

[1]  Zhou K, Huang X, Li Y, Zha D, Chen R, Hu X. Towards deeper graph neural networks with differentiable group normalization. arXiv preprint arXiv:2006.06972. 2020 Jun 12.

[2]  Shchur O, Mumme M, Bojchevski A, Günnemann S. Pitfalls of graph neural network evaluation. arXiv preprint arXiv:1811.05868. 2018 Nov 14.

Sincerely,

Paper2003 Authors

---

### Author Response · Authors · 2021-11-29
**Summary of updates from Authors**

Dear Reviewers and AC panel,

Thank you again for your valuable reviews that have helped improve and revise our submission. We are happy that the **theoretical novelty** of studying the over-smoothing problem of deep GNN through Graph Neural Tangent Kernel has been recognized by all five reviewers.  On the other hand, the **experimental contributions** have been recognized by reviewers Bi6F, CDt6, and sX4L.

We realize that the end of the rebuttal period is only one day away, and we would love to hear from the reviewers regarding our individual responses. We summarize the key points of our responses for a fast understanding:

(a) We have improved the explanation and discussion of the theorem, especially the discussion about the coefficients of the convergence rate, see the main context below Theorem 2 and Appendix C.1.

(b) We provide more experiments (Appendix F.4) and discussions (Appendix D) to further clarify the impact of C-DropEdge on the trainability of the Graph.

(c) We have introduced a new baseline method, which is a normalization-based approach [1] for comparison, and our method can outperform this method regarding different depths. More details can be found in Table 1.

(d) We have introduced a new dataset named Physics [2] which is a dense graph (average degree is 7.2 and edge keep rate is 6.96%). The experiments results (Table 5) show that C-DropEdge can consistently outperform the DropEdge method in this dataset.

We sincerely hope in keeping a positive discussion on the contributions of our work. Please do not hesitate to contact us if there are any more concerns that we could address. Thanks!

**References**

[1] Zhou K, Huang X, Li Y, Zha D, Chen R, Hu X. Towards deeper graph neural networks with differentiable group normalization. arXiv preprint arXiv:2006.06972. 2020 Jun 12.

[2] Shchur O, Mumme M, Bojchevski A, Günnemann S. Pitfalls of graph neural network evaluation. arXiv preprint arXiv:1811.05868. 2018 Nov 14.

Sincerely,

Paper2003 Authors

---

### Public Comment · ~Weilin_Cong1 · 2022-01-31
**Related work**

Dear authors,

Thanks for this interesting new perspective!

I would like to point out our related work in NeurIPS 2021: "On Provable Benefits of Depth in Training Graph Convolutional Networks”
 (https://openreview.net/forum?id=r-oRRT-ElX), in which we had a different opinion on over-smoothing to this paper.

In particular, we show that by removing dropout and weight decay, one can train deep GCNs with very high training accuracy (https://github.com/CongWeilin/DGCN/blob/master/DGL_code.ipynb), which implies over-smoothing might not happen in practice.

Besides, we also empirically show in Appendix E.3 that dropout is decreasing the training accuracy but reducing the generalization gap (difference between validation loss to training loss). Since the training accuracy is affected by dropout, it implies dropout is not solving “over-smoothing” (if it really exists).

It would be great if the authors could provide some connection to this paper, we believe this could be very beneficial for future studies.

Best,

Weilin

---

> ### Public Comment · ~Wei_Huang6 · 2022-02-03
> **Thank you for your interest**
>
> Dear Weilin,
>
> Thank you for taking an interest in our paper, I am happy to have further discussions with you.
>
> I like your work, which studies the trainability and generalization of deep GCNs. Understanding the GCN through a generalization perspective is of great interest, and I like it very much.
>
> I notice that the depth in your experiments is around 10. If I understand correctly, at this depth, we can still obtain low training error or high training accuracy, as confirmed by Fig 2 in our manuscript.
>
> On the other hand, I think a large graph with many components may not suffer from an over-smoothing problem even trained by deep GCNs. I am wondering if this is studied in your work.
>
> Best,
>
> Wei

---

### Decision · Program_Chairs · 2022-01-20

**Decision:**

Accept (Poster)

**Comment:**

In this paper, the authors established interesting theoretical results regarding the behavior Graph Neural Tangent Kernel (GNTK). They also provide sufficient evidence (some of which during rebuttal) that their approach is valid. We have had many discussions and I suggest that the authors apply reviewers' comments to the final version of their paper.